# TRPM7 kinase-mediated immunomodulation in macrophage plays a central role in magnesium ion-induced bone regeneration

Wei Qiao [1,2,3], Karen H. M. Wong[1,2], Jie Shen[1,2], Wenhao Wang[1,2], Jun Wu[1,2], Jinhua Li [1,2,4], Zhengjie Lin[1,2], Zetao Chen[5,6], Jukka P. Matinlinna[3], Yufeng Zheng [7], Shuilin Wu[8], Xuanyong Liu[9], Keng Po Lai [10,11], Zhuofan Chen [6✉], Yun Wah Lam[11✉], Kenneth M. C. Cheung[1,2] & Kelvin W. K. Yeung [1,2,12✉]

Despite the widespread observations on the osteogenic effects of magnesium ion ($Mg^{2+}$), the diverse roles of $Mg^{2+}$ during bone healing have not been systematically dissected. Here, we reveal a previously unknown, biphasic mode of action of $Mg^{2+}$ in bone repair. During the early inflammation phase, $Mg^{2+}$ contributes to an upregulated expression of transient receptor potential cation channel member 7 (TRPM7), and a TRPM7-dependent influx of $Mg^{2+}$ in the monocyte-macrophage lineage, resulting in the cleavage and nuclear accumulation of TRPM7-cleaved kinase fragments (M7CKs). This then triggers the phosphorylation of Histone H3 at serine 10, in a TRPM7-dependent manner at the promoters of inflammatory cytokines, leading to the formation of a pro-osteogenic immune microenvironment. In the later remodeling phase, however, the continued exposure of $Mg^{2+}$ not only lead to the over-activation of NF-κB signaling in macrophages and increased number of osteoclastic-like cells but also decelerates bone maturation through the suppression of hydroxyapatite precipitation. Thus, the negative effects of $Mg^{2+}$ on osteogenesis can override the initial pro-osteogenic benefits of $Mg^{2+}$. Taken together, this study establishes a paradigm shift in the understanding of the diverse and multifaceted roles of $Mg^{2+}$ in bone healing.

[1] Department of Orthopaedics and Traumatology, Li Ka Shing Faculty of Medicine, The University of Hong Kong, Hong Kong SAR, China. [2] Shenzhen Key Laboratory for Innovative Technology in Orthopaedic Trauma, The University of Hong Kong-Shenzhen Hospital, Shenzhen, China. [3] Dental Materials Science, Applied Oral Sciences, Faculty of Dentistry, The University of Hong Kong, Hong Kong SAR., China. [4] Centre for Translational Bone, Joint and Soft Tissue Research, University Hospital and Faculty of Medicine Carl Gustav Carus, Technische Universität Dresden, Dresden, Germany. [5] Department of Oral Implantology, Hospital of Stomatology, Guanghua School of Stomatology, Institute of Stomatological Research, Sun Yat-sen University, Guangzhou, China. [6] Zhujiang New Town Clinic, Hospital of Stomatology, Sun Yat-sen University, Guangzhou, China. [7] State Key Laboratory for Turbulence and Complex System and Department of Materials Science and Engineering, College of Engineering, Peking University, Beijing, China. [8] School of Materials Science and Engineering, Tianjin University, Tianjin, China. [9] State Key Laboratory of High Performance Ceramics and Superfine Microstructure, Shanghai Institute of Ceramics, Chinese Academy of Sciences, Shanghai, China. [10] Guangxi Key Laboratory of Tumor Immunology and Microenvironmental Regulation, Guilin Medical University, Guilin, China. [11] Department of Chemistry, City University of Hong Kong, Kowloon Tong, Hong Kong SAR, China. [12] China Orthopedic Regenerative Medicine Group (CORMed), Hangzhou, China. ✉email: chzhuof@mail.sysu.edu.cn; yunwlam@cityu.edu.hk; wkkyeung@hku.hk

Bone tissue has a substantial capacity for repair and regeneration after injury or surgical treatment. However, the natural healing of bone can be a slow process that often fails to restore the bone to its original strength and structure[1]. Thus, clinical interventions using orthopedic biomaterials are often required to accelerate bone healing while maintaining the amount and quality of bone mass. Magnesium ion ($Mg^{2+}$) is integral to bone homeostasis and metabolism. Deficiency in $Mg^{2+}$ is known to disrupt systemic bone metabolism, characterized by the inadequate bone formation and deregulated bone resorption[2–8]. In contrast, $Mg^{2+}$ supplement is beneficial to patients with osteoporosis[9]. Magnesium, its alloys[10–13] and derivatives[14–16] have been extensively studied as replacements of non-degradable metallic implants e.g., titanium alloys in bone surgeries. $Mg^{2+}$ modified biomaterials have shown a superior osteogenic capacity in many reports[14–21]. However, detrimental effects of $Mg^{2+}$ released upon degradation have also been observed[22–24]. These conflicting results may reflect the incomplete understanding of the roles of $Mg^{2+}$ in the complex biological process of bone healing.

Our group has recently reported that the incorporation of $Mg^{2+}$ in polycaprolactone (PCL) implant[18] and poly(lactic-co-glycolic acid) (PLGA) microsphere[19] can promote bone formation in a rat femoral defect model. We identified ~50–200 ppm as the optimal $Mg^{2+}$ concentration for promoting osteogenic activities of osteoblasts in vitro, as well as new bone formation in vivo[13,18]. Moreover, by using customized biomaterials that enable the controlled release of $Mg^{2+}$ at different stages of bone healing, we demonstrated that the bone regeneration rate and the quality of newly formed bone tissues depend on the release profile of $Mg^{2+}$ [18,19]. Bone healing is a complex process that involves the precise coordination of osteoclastogenesis and osteogenesis, through the interplay of multiple types of cells in a dynamic microenvironment. It is possible that the different cell types involved in various phases of bone healing, from early inflammation to the later bone formation and remodeling, may respond to $Mg^{2+}$ in different ways.

The monocyte-macrophage cell lineage has been recognized as a major player in bone regeneration and in acute inflammation responses to biomaterials, mainly due to their high plasticity in response to environmental cues and their multiple roles in bone homeostasis. According to their distinct functional properties, surface markers, and inducers, macrophages are characterized into several phenotypes (i.e., M1, M2a, M2b, and M2c)[25]. The pro-inflammatory cytokines caused by $Mg^{2+}$ deficiency can contribute to osteoclastogenesis[26,27], whereas the $Mg^{2+}$-induced anti-inflammatory cytokines and tissue repair factors benefit tissue regeneration[28,29]. The doping of $Mg^{2+}$ into titanium[30] and calcium phosphate cement[31] was demonstrated to promote the M2 polarization of macrophages. However, there is yet a consensus on which macrophage phenotype is more beneficial to bone regeneration because both M1[32,33] and M2[34,35] phenotypes have been reported to contribute to osteogenesis. Moreover, the conventional M1/M2 classification of macrophages has been challenged by a more heterogeneous grouping method, which suggests there may exist a continuum between M1 and M2 phenotypes yet to be identified[36]. Thus, the complexity of the $Mg^{2+}$-induced immunomodulation on macrophages, as well as its specific effects on the bone healing process in the complicated in vivo scenario requires further investigation.

In this study, we systematically analyzed the dose-dependent and time-dependent effects of $Mg^{2+}$ on the monocyte-macrophage-osteoblast axis in bone healing and investigated the underlying mechanisms behind the action of $Mg^{2+}$. We demonstrated a previously unknown immunomodulatory role of $Mg^{2+}$ at the early phase of bone healing, in which macrophages are stimulated, through a signaling pathway that involves transient receptor potential cation channel member 7 (TRPM7), to generate a specific pro-osteogenic immune microenvironment. At the later bone repair/remodeling phase, the prolonged presence of $Mg^{2+}$ impacts bone healing in another way, through activation of NF-κB signaling and inhibition on mineralization of extracellular matrix. We believe that these results will inspire the development of next-generation of $Mg^{2+}$-based degradable biomaterials for clinical uses that can better harness the healing power of $Mg^{2+}$.

## Results

**The time-dependent effect of $Mg^{2+}$ on bone regeneration.** To elucidate the time-dependent effect of $Mg^{2+}$ on bone healing, we developed an alginate-based hydrogel that allows a transient release of $MgCl_2$, at a concentration of ~10 mM, over one week (Fig. S1a). Critical-sized tunnel defects with a diameter of 2 mm were created in the distal end of rat femora[18,19]. $Mg^{2+}$ hydrogel was injected into the cavities at different time points after the injury. Using scanning electron microscopy with energy dispersive X-Ray spectroscopy (SEM-EDX), we demonstrated that the $Mg^{2+}$ releasing hydrogel significantly increased the magnesium content while decreasing the calcium content in the defects on day 7 post-injury (Fig. S1c–e). Meanwhile, plasma $Mg^{2+}$ concentration remained unchanged (Fig. S1f), indicating that the implant caused a localized change in $Mg^{2+}$ level without affecting systemic magnesium homeostasis. The impact of this implant on chloride levels would be negligible, as the physiological chloride concentration is over 100 times higher than the $Mg^{2+}$ concentration[37]. The efficiency of $Mg^{2+}$-induced bone repair was monitored using micro-CT over 8 weeks, compared to a control group in which equivalent alginate without $Mg^{2+}$ was injected and a sham group in which the injured animals were not injected with any hydrogel. No observable healing was found in the sham group after 8 weeks (Fig. 1b, c, S2a, b). Transient exposure of $Mg^{2+}$ during the first week after injury (Fig. 1a, Regimen 1 described in "Methods" section) led to a 3-fold increase in the trabecular bone fraction (BV/TV), a 2.5-fold increase in trabecular number (Tb. N), a 2-fold increase in bone mineral density (BMD of TV and BMD of BV) and a 2-fold increase in trabecular thickness (Tb.Th), as compared to the control and the sham groups (Fig. 1b, c, S2a, b). Nanoindentation test showed $Mg^{2+}$ hydrogel promoted new bone formation without compromising the mechanical properties, as Young's modulus of the $Mg^{2+}$-induced newly formed bone at post-operation week 8 was comparable to the control and sham group (Fig. S1b). Histological assessments by H&E staining (Fig. 1d, S2S2c) confirmed the increase in bone formation in the defect grafted with $Mg^{2+}$ releasing hydrogel. Meanwhile, the number of osteocalcins (OCN) positive osteoblasts increased in the $Mg^{2+}$ treated group compared to the control and sham groups (Fig. 1e), whereas the number of tartaric acidic phosphatase (TRAP) positive osteoclasts in the $Mg^{2+}$ treated group on day 56 decreased (Fig. 1f). Goldner's trichrome staining (Fig. 1g) and Calcein/Xylenol labeling (Fig. 1h, S2d) demonstrated that new bone formation and mineral apposition were more active in the $Mg^{2+}$ treated group than in the control.

Despite a higher trabecular bone fraction and bone mineral density, the overall beneficial effects of $Mg^{2+}$ were significantly attenuated when the delivery of the $Mg^{2+}$, even at the same dose, was delayed to the second week (Regimen 2, Fig. 2a–c, S3a, b). There was no significant difference in bone morphology according to H&E staining (Fig. 2d, S3c). Indeed, the number of $OCN^+$ osteoblasts remains unchanged when the $Mg^{2+}$ releasing hydrogel was applied at the second week of bone healing (Fig. 2e, S3d), while the number of $TRAP^+$ osteoclasts was only slightly lower than the control (Fig. 2f). When $Mg^{2+}$ was continuously delivered over the first two weeks post-injury (Regimen 3, Fig. 2g), the benefit of $Mg^{2+}$ hydrogel on bone formation measured by

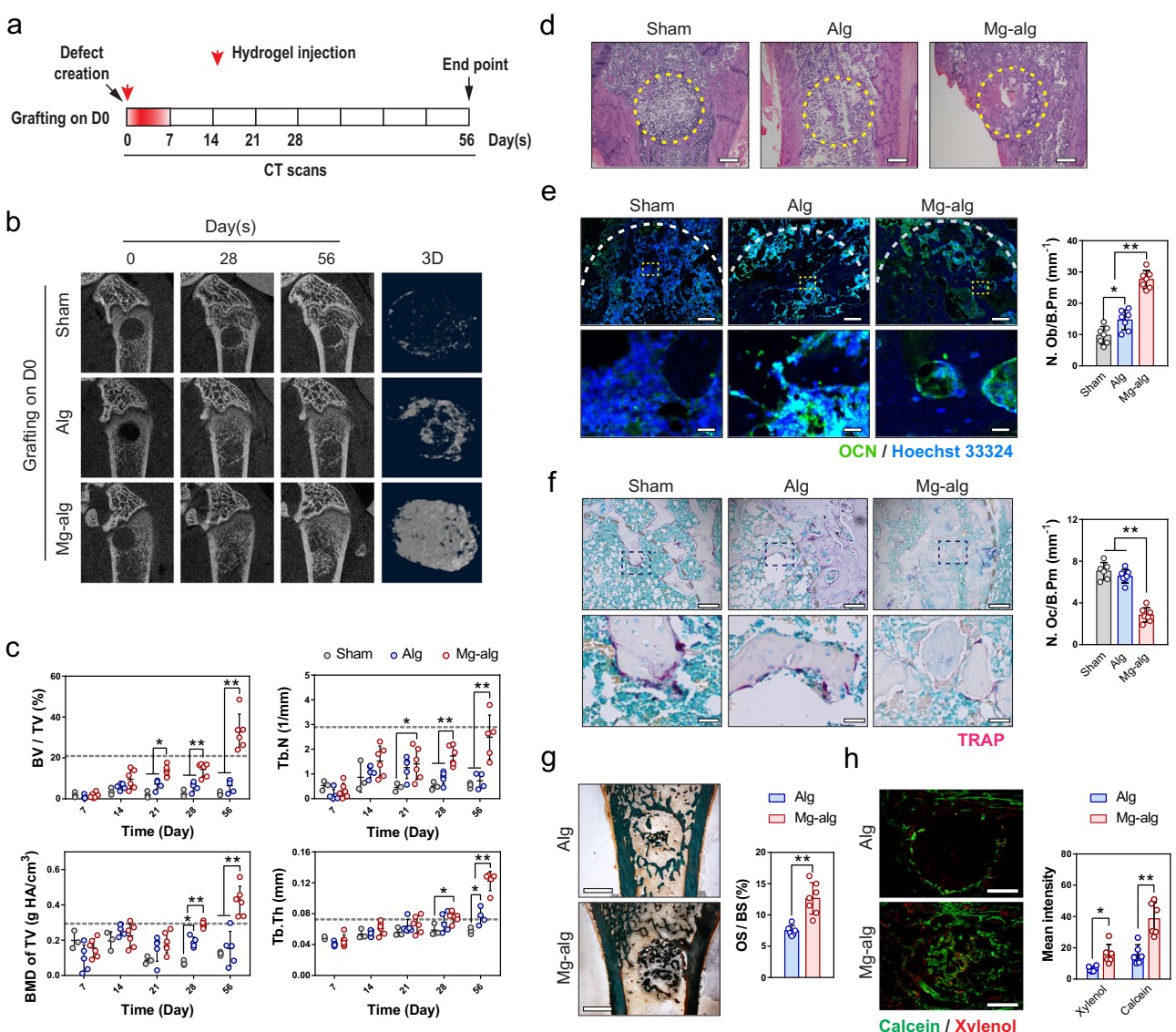

**Fig. 1 Mg$^{2+}$ releasing alginate promoted bone healing of defects in the rat femur. a** Mg$^{2+}$-crosslinked alginate was injected into the femur defect in rats right after the injury, hence the release of Mg$^{2+}$ was limited to the first week of the injury. **b** Representative micro-CT images and reconstructed 3D images of the defects in rat femora without grafting (Sham, $n = 3$), grafted with pure alginate (Alg, $n = 5$) or Mg$^{2+}$ releasing alginate (Mg-Alg, $n = 6$). **c** Corresponding measurements of trabecular bone fraction (BV/TV), trabecular number (Tb.N), bone mineral density (BMD of TV), and trabecular thickness (Tb.Th) showing the healing process of rat femoral defects. Sham, $n = 3$; Alg, $n = 5$; Mg-alg, $n = 6$. The dashed line shows the mean value of each bone parameter at the defect area before the operation. **d** Representative H&E staining images of the grafted defects in the rat femora, scale bars = 500 μm. **e** Representative immunofluorescent images and quantification showing the presence of osteoblasts (N.Ob/B.Pm, $n = 6$) in the grafted defects in the rat femora on day 56. Lower images (scale bars = 50 μm) are high-resolution versions of the boxed regions in the upper images (scale bars = 500 μm). **f** Representative TRAP staining images showing the presence of osteoclasts in the grafted defects in the rat femora, and histomorphological analysis of osteoclast numbers (N.Oc/B.Pm, $n = 6$) in the defects of femora on day 56. Lower images (scale bars = 40 μm) are high-resolution versions of the boxed regions in the upper images (scale bars = 200 μm). **g** Representative Goldner's trichrome staining of the grafted defects on day 56, scale bars = 1 mm, and quantitative analysis of osteoid surface per bone surface (OS/BS, $n = 6$) in the grafted femoral defects. **h** Representative images of calcein/xylenol labeling for bone regeneration in the rat femoral defects grafted with alginate or Mg$^{2+}$ releasing alginate, scale bars = 1 mm, and quantitative analysis of fluorescence intensity of calcein/xylenol ($n = 6$). Data are mean ± s.d. n.s. $P > 0.05$, *$P < 0.05$, **$P < 0.01$ by two-way ANOVA with Tukey's post hoc test (**c**), one-way ANOVA with Tukey's post hoc test (**e**, **f**), or Student's $T$-test (**g**, **h**).

micro-CT became negligible (Fig. 2h, i, Fig. S3e, f). In addition, bone morphology, as well as the number of OCN$^+$ osteoblasts and TRAP$^+$ osteoclasts were not different between the Mg$^{2+}$ treated group and the control (Fig. 2j–l, S2g, h). These observations demonstrate that Mg$^{2+}$ promotes bone healing only when delivered during the initial phase of repair, and a prolonged treatment resulted in unexpectedly detrimental effects on bone formation.

**Mg$^{2+}$-induced bone regeneration is mediated by macrophage activities.** As the effective window of Mg$^{2+}$ exposure coincided with the initial phase of inflammation dominated by macrophages[38], it is possible that the effect of Mg$^{2+}$ on bone formation might be mediated through its modulation of macrophages. Mg$^{2+}$ releasing hydrogel contributed to a significant increase in the number of CD68 positive cells in the proximity of the defect on day 7 after the operation (Fig. 3b and Fig. S4b, c). These cells likely

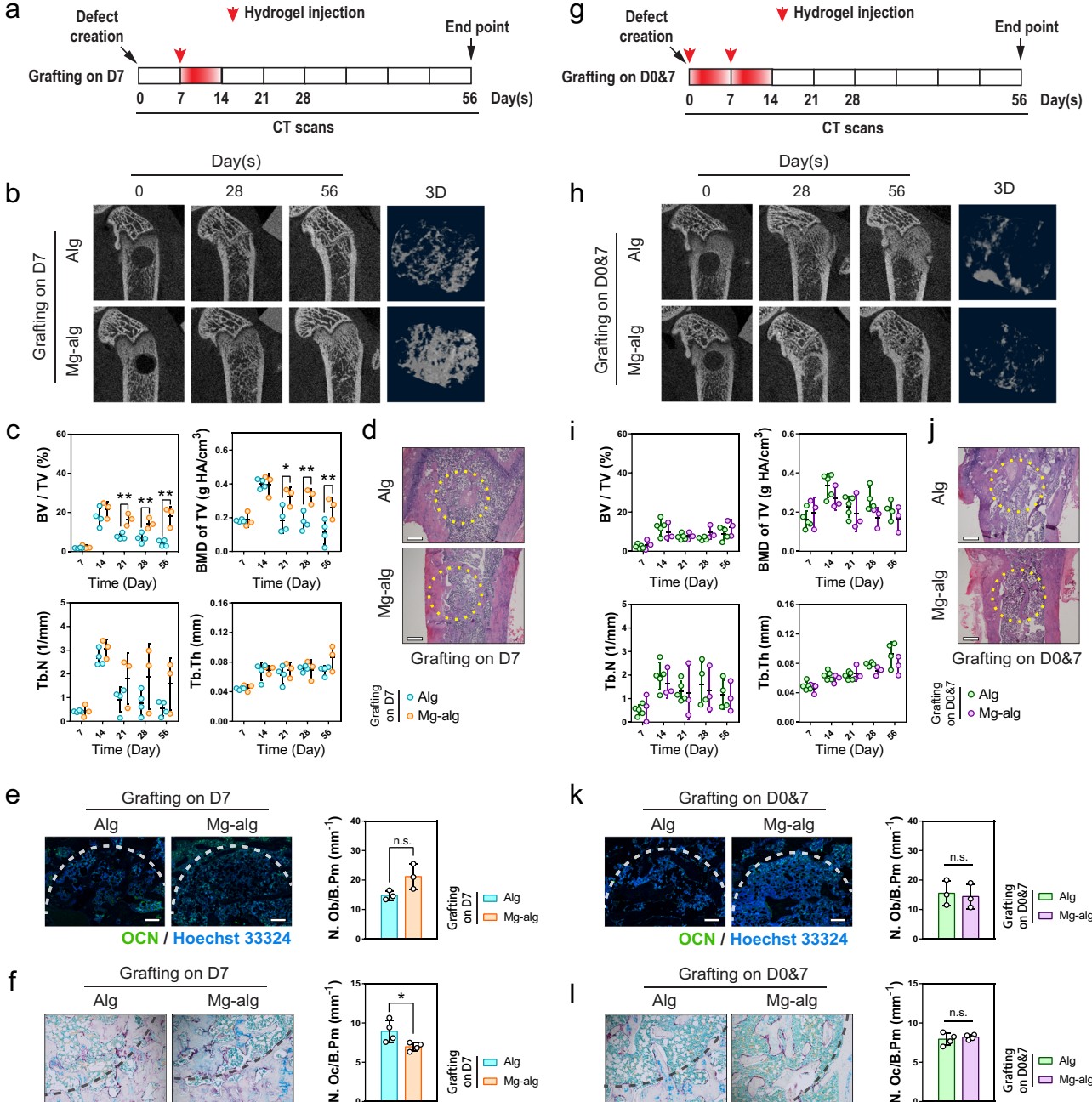

**Fig. 2 Delayed or prolonged delivery of Mg$^{2+}$ compromised its effects on bone healing. a** Mg$^{2+}$-crosslinked alginate was injected into the femur defect on the seventh day after the injury to exclude the effects of Mg$^{2+}$ on early phrase inflammation. **b** Representative micro-CT and reconstructed 3D images of the defects in rat femur on day 56 when the grafting was delayed. **c** Corresponding measurements of BV/TV, Tb.N, BMD of TV, and Tb.Th showing the healing process of rat femoral defects. Alg, $n = 4$; Mg-alg, $n = 3$. **d** Representative H&E staining images of the grafted defects in the rat femora, scale bars = 500 μm. **e** Representative immunofluorescent images and quantification showing the presence of osteoblast in the grafted defects in the rat femora on day 56, scale bars = 500 μm. **f** Representative TRAP staining images and quantification showing the presence of osteoclasts in the grafted defects in the rat femora on day 56 ($n = 4$), scale bars = 200 μm. **g** Mg-crosslinked alginate was injected into the femur defect in rats at both the first and seventh days after the injury to allow sustained release of Mg$^{2+}$ in the first two weeks of injury. **h** Representative micro-CT and reconstructed 3D images of the defects in rat femur on day 56 when the grafting was repeated ($n = 4$). **i** Corresponding measurements of BV/TV, Tb.N, BMD of TV and Tb.Th showing the healing process of rat femoral. Alg, $n = 4$; Mg-alg, $n = 3$. Data are mean ± s.d. **j** Representative H&E staining images of the grafted defects in the rat femora on day 56, scale bars = 500 μm. **k** Representative immunofluorescent images and quantification showing the presence of osteoblast in the grafted defects in the rat femora on day 56 ($n = 4$), scale bars = 500 μm. **l** Representative TRAP staining images showing the presence of osteoclasts in the grafted defects in the rat femora on day 56 ($n = 4$), scale bars = 200 μm. Data are mean ± s.d. n.s. $P > 0.05$, *$P < 0.05$, **$P < 0.01$ by two-way ANOVA with Tukey's post hoc test (**c**, **i**) or Student's $T$-test (**e**, **f**, **k**, **l**).

represented macrophages as the Mg$^{2+}$-dependent recruitment of CD68$^+$ cells vanished in animals treated with intraperitoneal administrations of liposome-encapsulated clodronate[39] (Fig. 3a, b). Meanwhile, the addition of Mg$^{2+}$ also contributed to a group of

TRAP$^+$ osteoclastic-like cells at the early stage (i.e., day 7) of bone healing (Fig. 3c). They are closely related to the group of macrophages responding to the stimulation of Mg$^{2+}$ because these CD68$^+$ osteoclastic-like cells (Fig. S4d) were also missing in the

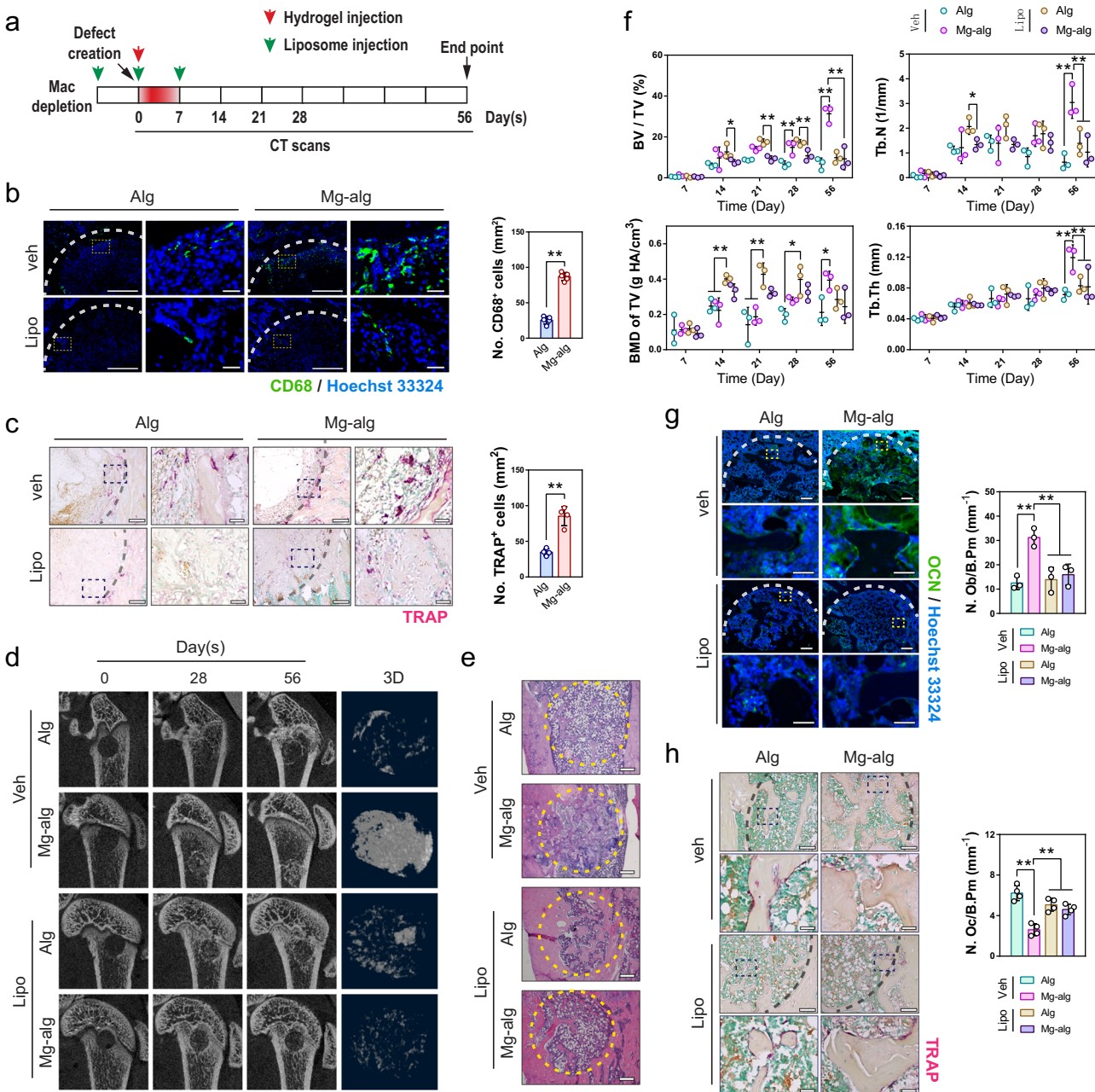

**Fig. 3 The key role of macrophages in Mg$^{2+}$-induced new bone formation. a** Mg-crosslinked alginate was injected into the femur defect in rats when their macrophages were selectively depleted by intraperitoneal administrations of liposome-encapsulated clodronate. **b** Representative immunofluorescent images showing the infiltration of CD68$^+$ macrophages on day 7 in the grafted defects in the rat femora and corresponding quantification for the number of CD68$^+$ macrophages ($n = 3$), right images (scale bars = 20 μm) are high-resolution versions of the boxed regions in the left images (scale bars = 200 μm). **c** Representative TRAP staining images showing the presence of TRAP$^+$ cells in the grafted defects in the rat femora on day 7 and corresponding quantification for the number of TRAP$^+$ cells ($n = 3$), right images (scale bars = 40 μm) are high-resolution versions of the boxed regions in the left images (scale bars = 200 μm). **d** Representative micro-CT and reconstructed 3D images of the defects in rat femur on day 56 ($n = 3$). **e** Representative H&E staining images of the grafted defects in the rat femora, scale bars = 500 μm. **f** Corresponding measurements of BV/TV, Tb.N, BMD of TV and Tb.Th showing the healing process of rat femoral defects ($n = 3$). **g** Representative immunofluorescent images showing the presence of osteoblasts in the grafted defects in the rat femora on day 56 ($n = 3$), lower images (scale bars = 100 μm) are high-resolution versions of the boxed regions in the upper images (scale bars = 500 μm). **h** Representative TRAP staining images showing the presence of osteoclasts in the grafted defects in the rat femora on day 56 ($n = 3$), lower images (scale bars = 40 μm) are high-resolution versions of the boxed regions in the upper images (scale bars = 200 μm). Data are mean ± s.d. n.s. $P > 0.05$, *$P < 0.05$, **$P < 0.01$ by Student's $T$-test (**b**, **c**), two-way ANOVA with Tukey's post hoc test (**f**), or one-way ANOVA with Tukey's post hoc test (**g**, **h**).

macrophage-depleted model (Fig. 3c). Compared with the vehicle group, which showed Mg$^{2+}$-induced new bone formation, the osteo-promoting effects of Mg$^{2+}$ were abolished in macrophage-depleted rats even when Mg$^{2+}$ was delivered in the optimal time

window. Indeed, Mg$^{2+}$ exposure appeared to delay bone healing in these animals: the BV/TV was lower in Mg$^{2+}$ treated group from day 14 to 28 relative to the control group (Fig. 3d, f, S4a), while histological analysis demonstrated that the effects of Mg$^{2+}$ on

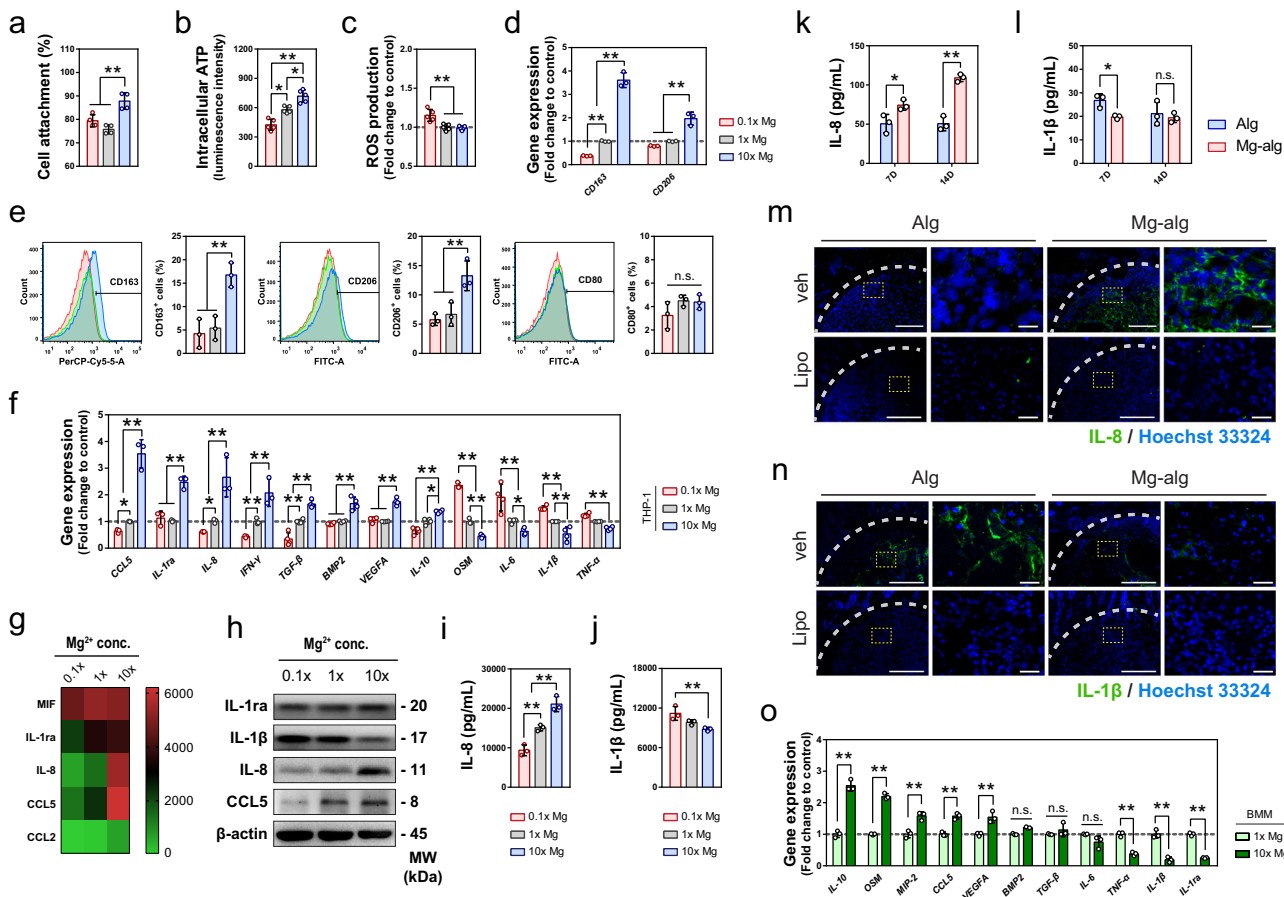

**Fig. 4 Mg$^{2+}$ regulated the inflammatory microenvironment through the immunomodulation of macrophages. a, b, c** The effects of different concentrations of Mg$^{2+}$ on the cell attachment (**a**, $n = 4$), intracellular ATP level (**b**, $n = 5$), and ROS production (**c**, $n = 4$) of macrophages differentiated from suspension THP1 monocytes. The data for cell attachment was expressed as a percentage of initially seeded THP-1 cells. **d** The effect of different concentrations of Mg$^{2+}$ on the gene expression of *CD163* and *CD206* in THP1-derived macrophages as evaluated by RT-qPCR ($n = 3$). **e** The effect of different concentrations of Mg$^{2+}$ on the polarization of macrophages was evaluated by the expression of CD163, CD206, and CD80 using flow cytometry ($n = 3$). **f** The relative expression of inflammatory-related genes regulated by the stimulation of Mg$^{2+}$ in THP1-derived macrophages ($n = 3$). **g** Major cytokines that respond to the stimulation of Mg$^{2+}$ determined by cytokine arrays were shown in a heat map. **h** Representative western blots showing the expression of IL-1ra, IL-1β, IL-8, and CCL5 of THP1-derived macrophages cultured in a medium supplemented with different concentrations of Mg$^{2+}$. **i, j** ELISA analysis showing the concentration-dependent effect of Mg$^{2+}$ on the production of IL-8 (**i**) and IL-1β (**j**) in THP1-derived macrophages ($n = 3$). **k, l** ELISA analysis on IL-8 (**k**) and IL-1β (**l**) in the grafted defects in the rat femora on day 7 after the operation ($n = 3$). **m, n** Representative immunofluorescent images showing the expression of IL-8 (**m**) and IL-1β (**n**) on day 7 in the grafted defects in the rat femora, ($n = 3$), right images (scale bars = 20 μm) are high-resolution versions of the boxed regions in the left images (scale bars = 200 μm). **o** The inflammatory-related genes regulated by the stimulation of Mg$^{2+}$ in mouse primary bone marrow macrophages (BMM, $n = 3$). Data are mean ± s.d. n.s. $P > 0.05$, *$P < 0.05$, **$P < 0.01$ by one-way ANOVA with Tukey's post hoc test (**a–e**, **i**, **j**) or two-way ANOVA with Tukey's post hoc test (**f**, **k**, **l**, **o**).

promoting osteogenesis while suppressing osteoclastogenesis became insignificant (Fig. 3e, g, h, S4b).

We conducted in vitro experiments to further delineate the role of Mg$^{2+}$ in macrophage functions by exposing THP1, a human monocyte cell line that can be differentiated into macrophages, to different Mg$^{2+}$ concentrations (named 0.1×, 1×, and 10× to represent the Mg$^{2+}$ concentrations in the respective medium relative to that in a physiological Mg$^{2+}$ level in the culture medium, 0.8 mM). An increase of Mg$^{2+}$ concentration in the culture medium significantly promoted the maturation of suspension monocytes into adhered macrophages (Fig. 4a, S5a). Mg$^{2+}$ also increased the activity of THP1-derived macrophages, as evidenced by the increase of intracellular ATP levels (Fig. 4b) and the number of mitochondria (Fig. S5b). Furthermore, Mg$^{2+}$ deficiency enhanced ROS production (Fig. 4c). RT-qPCR demonstrated that 10× Mg$^{2+}$ treatment resulted in an increased expression of M2 macrophage surface markers CD163 and CD206 (Fig. 4d). This was further supported by flow cytometry

data showing that the number of macrophages expressing M2 surface markers CD163 and CD206 was increased by the stimulation of 10× Mg$^{2+}$, while the number of M1 macrophages characterized by the expression of CD80 remained unchanged (Fig. 4e). Our RT-qPCR data also revealed the effect of 10× Mg$^{2+}$ on upregulating a series of genes encoding cytokines favoring osteogenesis, such as CCL5, IL-1ra, IL-8, TGF-β1, BMP2, VEGFA, IL-10, while downregulating genes encoding cytokines favoring osteoclastogenesis, including OSM, IL-6, IL-1β, TNF-α (Fig. 4f). Using cytokine array, the major cytokines secreted by macrophages upon the stimulation of Mg$^{2+}$ were found to be IL-1ra, IL-8, and CCL5, which were distinct from traditionally characterized M1 or M2 phenotypes (Fig. 4g, S5c). We confirmed the effects of Mg$^{2+}$ on the expression of IL-1ra, IL-8, CCL5, and IL-1β by western blots (Fig. 4h). We also demonstrated that the effects of Mg$^{2+}$ on the levels of IL-8 and IL-1β in THP1-derived macrophages to be both time-dependent and concentration-dependent (Fig. 4i, j S5d). Our findings on THP1-derived

macrophages were validated on primary mouse bone marrow macrophages (BMM). We found that the exposure of BMM to $10\times$ $Mg^{2+}$ contributed to similar immunomodulatory effects as observed in THP1-derived macrophages (Fig. 4o): genes upregulated by $10\times$ $Mg^{2+}$ included IL-10, OSM, CCL5, and VEGFA, while the two major pro-inflammatory genes, TNF-α and IL-1β, were downregulated. Due to the lack of IL-8 in mice[40], we confirmed the finding on the effect of $Mg^{2+}$ on IL-8 in mouse BMM by examining the expression of Macrophage Inflammatory Protein 2 (MIP-2), the murine structural and functional homolog of human IL-8[41,42]. We found that $10\times$ $Mg^{2+}$ treatment led to an ~1.5-fold increase in the expression of MIP-2 compared to the control. These in vitro findings were supported by our immunostaining and ELISA data in vivo showing that $Mg^{2+}$ released from the hydrogel led to a significant increase in the level of IL-8 (Fig. 4k, m, S4c), CCL5 (Fig. S5e), and IL-1ra (Fig. S5f), as well as a significant decrease in the level of IL-1β (Fig.4l, n, S4c) in the femoral defects one-week post-operation. However, in macrophage-depleted animals, the level of these cytokines remained low and unchanged regardless of the addition of $Mg^{2+}$ in the hydrogel. Taken together, our in vitro and in vivo data corroborate that an elevation of extracellular $Mg^{2+}$ triggered a change in cytokine expression in macrophages.

**The central role of TRPM7 in $Mg^{2+}$-induced inflammatory modulation in macrophages.** Using Mag-Fluo-4, an $Mg^{2+}$ specific dye, we observed a rapid increase of intracellular $Mg^{2+}$ levels in macrophage upon the addition of $Mg^{2+}$ in the culture medium, peaking at ~8 min before reaching a steady-state higher than the baseline (Fig. 5a, b, Fig. S6a). Our ICP-OES data also verified that the $Mg^{2+}$ level was consistently higher than the baseline when cultured in supplemented medium (Fig. S6b). As expected, when the channel activity of TRPM7, an $Mg^{2+}$ transporter, was inhibited by FTY720, a potent TRPM7 blocker that inhibits channel activity by reducing the open probability[43], the influx of $Mg^{2+}$ became insignificant (Fig. 5a, b). In addition, our RT-qPCR data showed that $10\times$ $Mg^{2+}$ contributed to a more than two-fold upregulation of TRPM7 expression in THP1-derived macrophages (Fig.5c) and in mouse BMM (Fig. 6c), relative to the marginal increase in the MagT1 gene (Fig. 5c). Using an antibody that targets the C-terminal of TRPM7, a region containing the TRPM7-cleaved kinase fragments (M7CKs)[44], we demonstrated that $Mg^{2+}$ treatment increased the overall level of full-length TRPM7 in THP1-derived macrophages, with a corresponding increase in M7CKs (Fig. 5e). This observation was also validated in vivo by the increased expression of TRPM7 at the proximity of the Mg-alg grafted defect compared with Alg grafted control (Fig. S6c, d). Immunofluorescence staining (Fig. 5d, Fig. 6d) and subcellular fractionation (Fig. 5f) confirmed the accumulation of M7CKs in macrophage nuclei upon the stimulation of $Mg^{2+}$. Moreover, the phosphorylation of Histone H3 at residue S10 (H3S10p) was found to be increased in THP1-derived macrophages (Fig. 5g, Fig. S6e) and mouse BMM (Fig. 6e) after $Mg^{2+}$ treatment, consistent with the role of M7CKs in H3 phosphorylation[44]. Using super-resolution microscopy, we demonstrated that nuclear M7CKs were localized in foci that overlapped with a subset of H3 foci (Fig. S6f). Chromatin immunoprecipitation (ChIP) assays in THP1-derived macrophage showed that $Mg^{2+}$ increased the association of H3S10p with the IL-8 promoter (Fig. 5h). Moreover, the gene loci at the proximity of the IκBα promoter were also enriched in pH3S10-containing chromatin in an $Mg^{2+}$-dependent manner (Fig. 6i).

To elucidate the role of TRPM7 in $Mg^{2+}$-mediated inflammation modulation, we selectively silenced the expression of TRPM7 with siRNA (Fig. 5j, S6g) or blocked the channel activity with

FTY720 (Fig. 5b). Consequently, the nuclear translocation of M7CKs and the phosphorylation of Histone H3 were inhibited (Fig. 5k, l). Moreover, the effect of $Mg^{2+}$ on cytokine expressions, such as the induction of IL-8 and the downregulation of IL-1β, was completely abolished (Fig. 5m, n). Interestingly, the loss of TRPM7 function appeared to disrupt the baseline expression of various inflammatory cytokines, such as IL-8, IL-1β, TNF-α, and IL-1ra. As TRPM7 transports many divalent metal cations (e.g., $Ca^{2+}$, $Zn^{2+}$, $Mn^{2+}$, and $Co^{2+}$) other than $Mg^{2+}$[45,46], it is possible that the inhibition of TRPM7 may alter macrophage functions in multiple ways.

**$Mg^{2+}$ stimulated early osteoclastic differentiation.** We demonstrated that prolonged exposure (i.e., 3 days or more) to $Mg^{2+}$ led to an increased expression of IKK-α and IKK-β in THP1-derived macrophages (Fig. 6f), which contributed to the phosphorylation of IκB and the nuclear translocation of p65 (Fig. 6f, S6h). This suggests the stimulation of $Mg^{2+}$ is associated with the activation of the NF-κB signaling pathway in macrophages. In addition, siRNA knockdown of TRPM7 and, to a lesser extent, the chemical blockage of TRPM7, downregulated the expression of IκB kinases (IKK-α/β), IκB phosphorylation, and nuclear p65 (Fig. 6g). In consistent with the role of NF-κB signaling pathway in osteoclastogenesis, our data showed that an extended exposure (i.e., 14 days or more) of $Mg^{2+}$ increased the number of TRAP+ cells (Fig. 6h, i), and extracellular TRAP activity (Fig. 6j) during the in vitro osteoclastic induction of THP1-derived macrophages by RANKL and M-CSF. However, our RT-qPCR data showed that only early osteoclastic markers (i.e., TRAP, RANK, and M-CSF) were elevated by the addition of $Mg^{2+}$, whereas the late osteoclastic markers (i.e., CTSK and CTR), which indicates the maturation of osteoclasts, were downregulated (Fig. 6k). Supporting this observation, $10\times$ $Mg^{2+}$ did not lead to significant phosphorylation of JNK and upregulation of nuclear factor-activated T cells c1 (NFATc1), which both play a crucial role in the terminal maturation of osteoclast (Fig. 6l). Also, our in vivo data showed a transient exposure to $Mg^{2+}$ resulted in more TRAP+ cells at the first week of bone healing but fewer TRAP+ cells on day 56 post-injury (Fig. 1f). Hence, $Mg^{2+}$ stimulated the NF-κB signaling pathway, in a TRPM7-dependent manner, and early osteoclastogenic markers, but there is no evidence to support that this early differentiation resulted in mature osteoclasts.

**$Mg^{2+}$-mediated cytokines from macrophage contribute to bone formation.** To demonstrate the effect of the macrophage-dependent inflammatory microenvironment on osteogenesis, we treated human mesenchymal stem cells (MSC) with conditioned media (CM) harvested from THP1-derived macrophages pre-incubated in different concentrations of $Mg^{2+}$ (named CM-0.1×, CM-1×, and CM-10×). The concentrations of $Mg^{2+}$ in all CM were adjusted to 1× before use (see "Methods" section). As expected, CM-10× (CM harvested from THP1-derived macrophages pre-incubated in $10\times$ $Mg^{2+}$) significantly increased the number (Fig. 6m) and osteogenic differentiation of MSC, as demonstrated by a two-fold increase in ALP activity (Fig. 6n) and the upregulation of osteogenic markers, including ALP, COL1A1, OPN, IBSP, BGLAP, and Osterix at both mRNA (Fig. 6o) and protein (Fig. S7a) levels. Meanwhile, CM-0.1× suppressed MSC proliferation and osteogenic gene expression (Fig. 6m–o). Alizarin red staining also revealed that CM from $Mg^{2+}$-treated macrophage culture could accelerate the formation of mineralized nodules in MSC (Figs. 7a, S8a). This effect was more pronounced when CM from $Mg^{2+}$ stimulated macrophage was administered during the first week of osteogenic induction of MSC (Fig. 7a).

Since our in vitro cytokine profiling (Fig. 4f–h) and in vivo immunostaining indicated IL-8 and IL-1β were two of the most

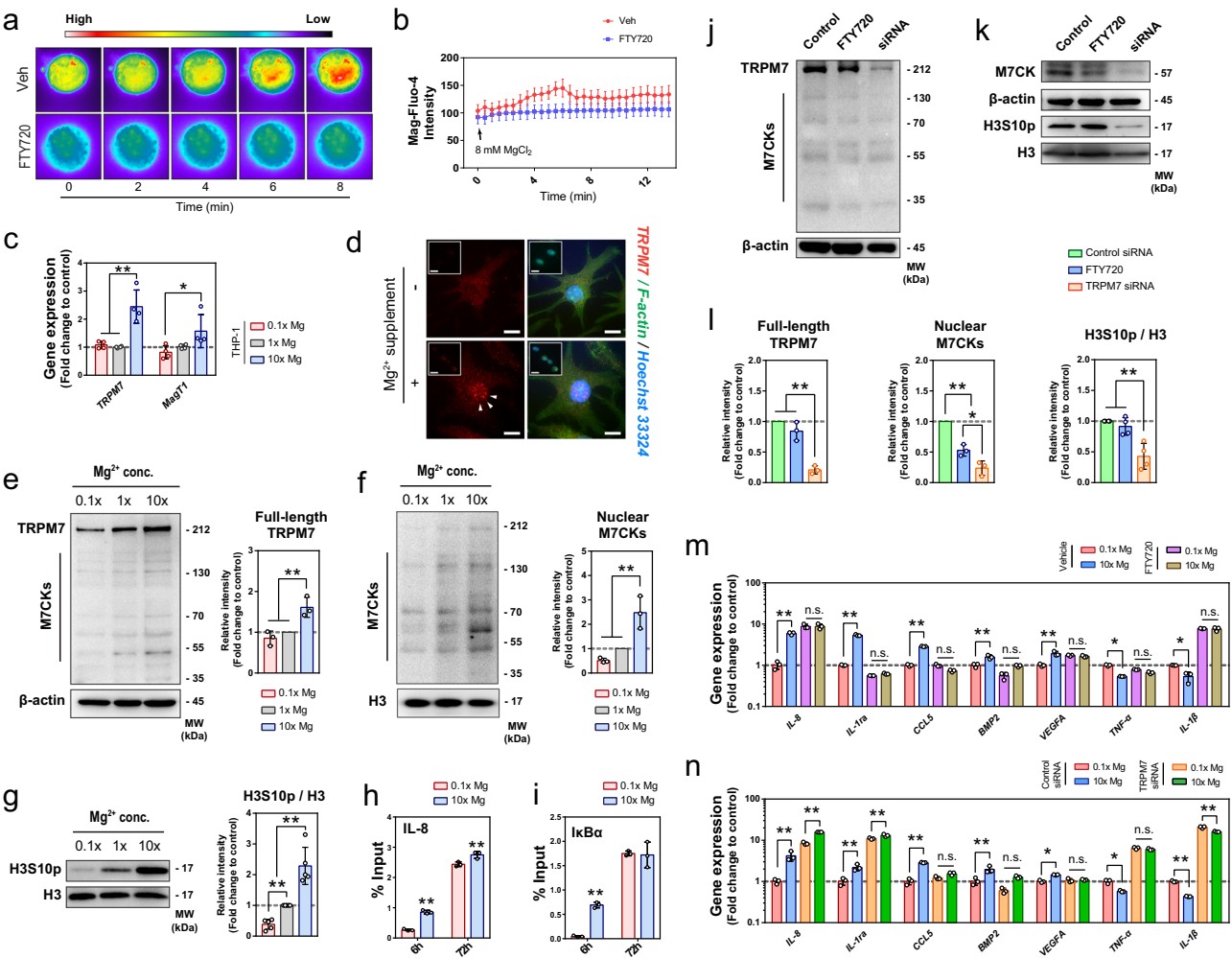

**Fig. 5 The involvement of TRPM7 in Mg²⁺-induced immunomodulation in macrophages. a, b** Representative fluorescence images (**a**) showing the influx of Mg²⁺ into THP1-derived macrophages after the addition of 8 mM MgCl₂. Color scale bar from low (black to blue) to high (red to white) indicates the level of Mg²⁺. **b** Time-course changes in intracellular Mg²⁺ quantified by measuring the intensity of fluorescence ($n = 5$). **c** The gene expression of *TRPM7* and *MagT1* in THP1-derived macrophages upon the stimulation of different concentrations of Mg²⁺ determined by RT-qPCR ($n = 4$). **d** Representative fluorescence images showing the nuclear accumulation of TRPM7 in THP1-derived macrophages after the stimulation of Mg²⁺. Inserts showed the staining with the cell permeabilization before the fixation to better demonstrate the nucleus bound TRPM7. Scale bars = 5 μm. **e** Representative western blots and the corresponding quantification showing the effects of Mg²⁺ on the expression of TRPM7 and its cleaved kinase fragments (M7CKs) in THP-1 derived macrophages ($n = 3$). **f, g** Western blots of nuclear proteins probed with anti-TRPM7 and the corresponding quantification showing the stimulation of Mg²⁺ contributed to an increased nuclear fraction of M7CK ($n = 3$) (**f**) and upregulated phosphorylation of Histone H3S10 ($n = 5$) (**g**). **h, i** CHIP assay showing the detection of promoters of IκBα (**h**) and IL-8 (**i**) in the immunoprecipitates of H3S10p was upregulated by the stimulation of Mg²⁺. (**j–l**) Representative western blots showing the effects of FTY720 and TRPM7 siRNA on the expression of TRPM7 (**j**, $n = 3$), its nuclear M7CKs (**k**, $n = 3$), and the phosphorylation of Histone H3S10 (**k**, $n = 4$) in THP1-derived macrophages, as well as corresponding quantifications (**l**). **m, n** The effects of FTY720 (**m**) and TRPM7 siRNA (**n**) on the inflammatory gene expression in THP1-derived macrophages ($n = 3$). Data are mean ± s.d. n.s. $P > 0.05$, *$P < 0.05$, **$P < 0.01$ by two-way ANOVA with Tukey's post hoc test. (**c**, **h**, **m**, **n**) or one-way ANOVA with Tukey's post hoc test (**e–g**, **l**).

significantly regulated cytokines from macrophages in response to Mg²⁺, we tested the effect of these two factors, individually and in combination, on MSC differentiation. The results indicated that human recombinant IL-8 replicated the effect of CM-10×, as evidenced by the significant increase in MSC proliferation (Fig. S7b), ALP activity (Fig. 7c), osteogenic genes expression (Fig. 7e), and mineralization (Fig. 7f). Although IL-1β led to a marginal increase in ALP activity and mineralization formation, it antagonized the osteogenic effects of IL-8 (Fig. 7c, f). Our data also showed that the osteogenic effect of recombinant IL-8, like CM-10×, was more prominent when it was added in the first week of culture. The presence of IL-8 in the late stage of osteogenic differentiation, instead, impaired the mineralization process (Fig. 7e). We next compared the osteogenic potential of CM-10× with and without the addition of

IL-8 neutralizing antibodies. IL-8 neutralizing antibody could abolish the osteogenic effect of CM-10× in MSC, as there was no significant difference in the ALP activity and osteogenic gene expression between MSC cultured in CM-1× and CM-10× in the presence of IL-8 neutralizing antibody (Fig. 7b, d). Interestingly, the effect of CM-10× on the mineralization of MSC was only abolished by IL-8 neutralizing antibody when it was added in the first week of culture. If the depletion of IL-8 was extended to the second week of culture, it tended to favor the mineralization (Fig. S7c). Together, these data suggest that Mg²⁺ treated macrophages secreted IL-8, which induced osteogenesis in MSC especially when administrated during the initial phase of differentiation.

To distinguish the effect of macrophage-derived molecules from the direct action of Mg²⁺ on MSC, we also evaluated the

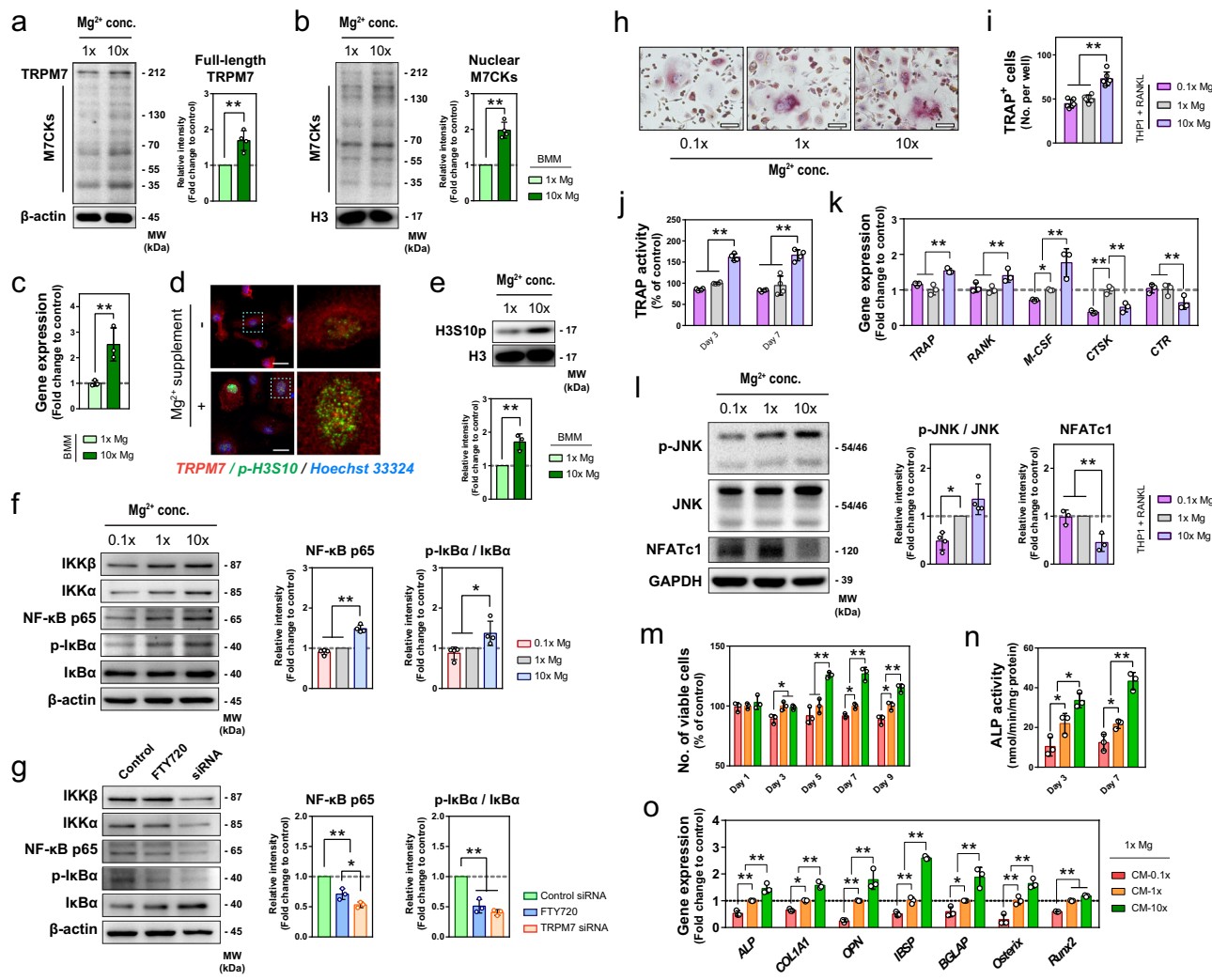

**Fig. 6 The effects of Mg²⁺ on macrophages and resulting impacts on osteoclastogenesis and osteogenesis. a, b** Representative western blots and the corresponding quantification showing the effects of Mg²⁺ on the expression of TRPM7 (**a**, $n = 4$) and its nuclear M7CKs (**b**, $n = 4$) in mouse BMM. **c** The gene expression of *TRPM7* in mouse BMM upon the stimulation of 10× Mg²⁺ determined by RT-qPCR ($n = 3$). **d** Representative fluorescence images showing the nuclear accumulation of TRPM7 and phosphorylation of Histone H3S10 in mouse BMM after the stimulation of Mg²⁺. Scale bars = 5 μm. right images are high-resolution versions of the boxed regions in the left images (scale bars = 20 μm). **e** Representative western blots and corresponding quantification showing Mg²⁺ upregulated the phosphorylation of Histone H3S10 in mouse BMM ($n = 3$). **f, g** Representative western blots and the corresponding quantification showing the concentration-dependent effect of Mg²⁺ (**g**) and the influence of FTY720 or TRPM7 siRNA on the activation of NF-κB signaling in THP1-derived macrophages ($n = 4$). **h, i** Representative microscopy images (**h**) and quantitative data (**i**) showing the effect of Mg²⁺ on the formation of multi-nuclear TRAP⁺ cells from THP1-derived macrophages stimulated by RANKL and M-CSF ($n = 6$). **j, k** The effect of Mg²⁺ on the extracellular TRAP activity (**j**, $n = 4$) and osteoclastic-related gene expression (**k**, $n = 3$) in THP1-derived macrophages stimulated by RANKL and M-CSF. **l** Representative western blots and the corresponding quantification showing the effects of Mg²⁺ on the phosphorylation of JNK ($n = 4$) and the expression of NFATc1 ($n = 3$). **m, n, o** The number of viable cells (**m**, $n = 3$), ALP activity (**n**, $n = 3$), and osteogenic related gene expression (**o**, $n = 3$) of MSC cultured in conditional medium from macrophages stimulated with different concentrations of Mg²⁺. Data are mean ± s.d. n.s. $P > 0.05$, *$P < 0.05$, **$P < 0.01$ by Student's T-test (**a**–**c**, **e**), one-way ANOVA with Tukey's post hoc test (**f**, **g**, **i**, **l**), or two-way ANOVA with Tukey's post hoc test (**j**, **k**, **m**, **n**, **o**).

osteogenic potential of MSC cultured in different levels of Mg²⁺. Mg²⁺ deficiency (0.1×) significantly reduced the ALP activity and the expression of osteogenic marker genes of MSC (Fig. 7g, S7e). Compared to CM-10×, which contained 1× Mg²⁺, an elevated level of extracellular Mg²⁺ (10×) only contributed to a < 10% increase in MSC proliferation (Fig. S7d) and a marginal increase in early osteogenic gene markers, such as *COL1A1, Osterix,* and *Runx2* (Fig. S7e). Instead, most of the osteogenic markers, such as *ALP, OPN,* and *BGLAP* were not affected by the addition of Mg²⁺ (Fig. S7e, f). Moreover, even as Sirius red staining showed that 10× Mg²⁺ promoted the formation of extracellular matrix (Fig. S8a), a prolonged (2 weeks) exposure of a high level of Mg²⁺ (10×) could suppress the spontaneous mineralization (Fig. 7h, S8a).

Interestingly, by exposing MSC to high (10×) Mg²⁺ in different time windows, we noticed that the inhibitory effect of high Mg²⁺ on MSC mineralization was only observed when it was administered during the second half of the two-week culture experiment (Fig. 7h). To further explore the temporal effects of Mg²⁺ on osteogenic differentiation, we analyzed precipitation formed under simulated physiological conditions using transmission electron microscopy (TEM). The typical TEM images and selected area electron diffraction (SAED) patterns of the precipitated calcium salts showed that the formation of crystallized apatite was significantly inhibited by the increase of Mg²⁺ level in the medium (Fig. S8b). X-ray diffraction data also indicated that the addition of Mg²⁺ led to a dramatic reduction of

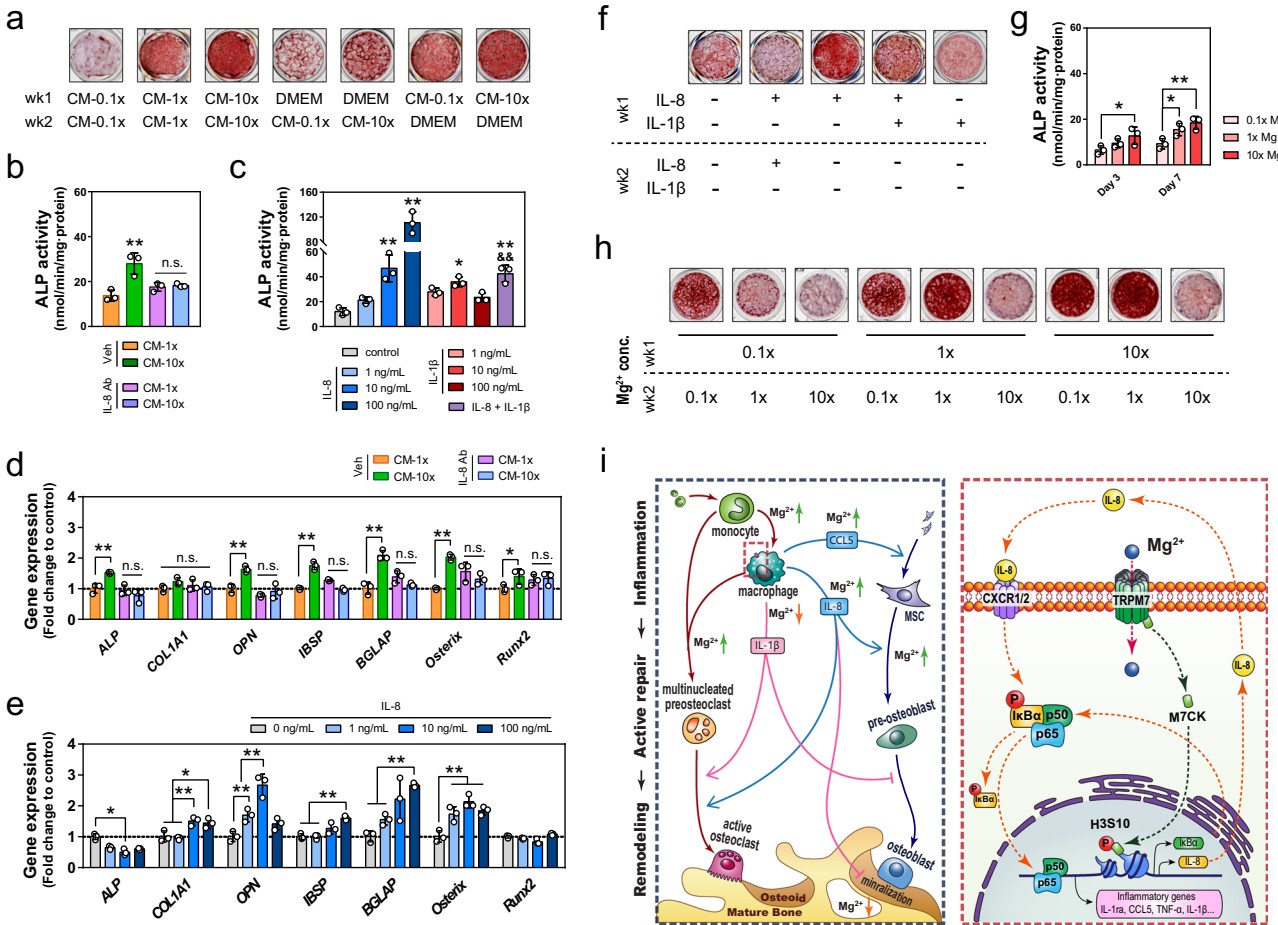

**Fig. 7 The effects of Mg$^{2+}$ and its modulated inflammatory microenvironment on osteogenesis. a** Alizarin Red staining of mineralized nodules of MSC treated with conditional medium from Mg$^{2+}$-treated macrophages at either early or late stage of osteogenic induction. **b** The ALP activity of MSC cultured in conditional medium from macrophages with or without the addition of IL-8 neutralizing antibody ($n = 3$). **c** The ALP activity of MSC cultured in medium supplemented with recombinant human IL-8 or IL-1β ($n = 3$). **d** The osteogenic-related gene expression of MSC cultured in conditional medium from macrophages with or without the addition of IL-8 neutralizing antibody ($n = 3$). **e** The osteogenic-related gene expression of MSC cultured in medium supplemented with different concentrations of recombinant human IL-8 ($n = 3$). **f** Alizarin Red staining of mineralized nodules of MSC treated with recombinant human IL-8 or IL-1β. **g** The ALP activity of MSC cultured in DMEM supplemented with different concentrations of Mg$^{2+}$ ($n = 3$). **h** Alizarin Red staining showing the mineralization of MSC treated with different concentrations of Mg$^{2+}$ at either early or late stage of osteogenic induction. **i** The schematic shows the mechanism in which Mg$^{2+}$ modulates both macrophages and mesenchymal stem cells in the bone healing process. Data are mean ± s.d. n.s. $P > 0.05$, *$P < 0.05$, **$P < 0.01$ by one-way ANOVA with Tukey's post hoc test (**b**, **c**) or two-way ANOVA with Tukey's post hoc test (**d**, **e**, **g**).

the hydroxyapatite phase, the predominant inorganic component in hard tissues (Fig. S8c).

## Discussion
Mg$^{2+}$ has been extensively used for the modification of orthopedics biomaterials due to observations that Mg$^{2+}$ can contribute to enhanced osteogenesis[18,47]. Previous studies have speculated that Mg$^{2+}$ modulates multiple signaling pathways at different stages in the differentiation process of mesenchymal stem cells toward osteoblast lineage[24,47–49]. For example, Mg$^{2+}$ is suggested to promote the proliferation and osteogenic differentiation of MSC by sequentially activating MAPK/ERK signaling pathway and Wnt/β-catenin signaling pathway[49]. Moreover, the activation of transforming growth factor-beta (TGF-β) and bone morphogenic protein (BMP) signaling, who plays fundamental roles in regulating MSC differentiation during skeletal development, is associated with the stimulation of Mg$^{2+}$[23]. Although our in vitro data in this study verified the direct osteogenic action of Mg$^{2+}$ on MSC, especially at the initiation of osteogenic differentiation, we noticed excessive Mg$^{2+}$ did not contribute to late osteoblastic activities, especially the

mineralization of the extracellular matrix. Indeed, previous studies suggest that the addition of Mg$^{2+}$ needs to be tailored to an optimal range (around 2–8 mM) to achieve its osteogenic effects, otherwise, the presence of high levels of Mg$^{2+}$ can be detrimental to the osteogenic differentiation of bone-forming cells[18,48,50]. Hence, the conventional approach that focuses on the effect of Mg$^{2+}$ on the osteoblast lineage cannot fully address the osteogenic effect of Mg and its alloy, especially in scenarios where bone-forming cells are continuously challenged by high concentrations of Mg$^{2+}$ over a long period[13,51]. Recently, implant-derived Mg$^{2+}$ is demonstrated to trigger the release of neuronal calcitonin gene-related polypeptide-α from dorsal root ganglia, resulting in enhanced osteogenic differentiation of periosteum-derived stem cells and improved bone-fracture healing[52]. This finding suggests that Mg$^{2+}$ may contribute to the osteogenic differentiation of osteoblast lineage through the activation of various other cell types in the bone microenvironment.

By grafting Mg$^{2+}$ releasing hydrogel at different stages of bone healing, our animal studies revealed an effective window for the administration of Mg$^{2+}$ to achieve bone healing: the initial inflammation stage is more important than the subsequent active

bone repair stage. As macrophages are known for their pivotal roles in the innate inflammatory response and their regulatory effects on bone homeostasis in tissue regeneration[38,53,54], we selectively depleted the phagocytes in the early inflammation stage using liposome clodronate and showed that the osteogenic effect of $Mg^{2+}$ was abolished. This suggests macrophages are the primary target cells responsible for $Mg^{2+}$-induced bone healing, which corroborates with numerous studies showing macrophages play important roles in the healing of bone tissues[55–58]. In this study, our in vivo and in vitro data have provided compelling evidence showing the cellular activities of macrophages, especially the cytokine release profiles, to be significantly affected by the presence of $Mg^{2+}$. First, the presence of $Mg^{2+}$ facilitates the recruitment and activation of monocyte towards matured macrophages. Interestingly, although the upregulation of CD163 and CD206 suggest an M2 polarization of the $Mg^{2+}$-treated macrophages, their cytokine secretion profile was distinct from the classical M1 and M2 regimes[59], as inflammatory genes typically observed in M1 macrophages (e.g., *IL-1β* and *TNF-α*) and in M2 macrophages (e.g., *IL-10*) were not significantly upregulated. Moreover, genes that are known to be involved in macrophage-mediated osteogenesis (e.g., *BMP2*[58], *OSM*[32], *TGF-β1*[60]) were not dramatically altered by $Mg^{2+}$ treatment. Instead, $Mg^{2+}$ triggers a previously unknown cytokine profile, characterized by an increased level of CCL5, IL-8, and IL-1ra and the reduction of IL-1β. CCL5 and IL-8 are traditionally recognized as pro-inflammatory factors responsible for cell recruitment to the sites of injuries and inflammation[61,62], however, both of them have been demonstrated capable of recruiting MSCs to bone healing site[63,64]. Moreover, IL-8 has been shown as a crucial mediator for angiogenesis[61,65], the commitment of MSCs to bone regeneration[63,66], and restricting bone resorption activity of osteoclasts[67]. IL-1β is one of the major pro-inflammatory cytokines that stimulate osteoclastogenesis[68]. A reduction of IL-1β in response to the addition of $Mg^{2+}$, possibly due to the anti-inflammatory effect of $Mg^{2+}$ and the IKK-dependent activation of NF-κB[69], contributes to the generation of new bone. In addition, $Mg^{2+}$ also upregulated the expression of IL-1 receptor antagonist (IL1-ra), which further limits IL-1β functions[70]. Collectively, our observations suggest that $Mg^{2+}$ stimulated macrophages to a cytokine mixture tailored for bone regeneration.

TRPM7 is a predominant $Mg^{2+}$ channel in mammalian cells. Conditional gene deletion experiments suggest that this transporter is involved in a variety of immune-related functions, including macrophage activation and polarization in a current dependent manner[71], but the implications of these functions in tissue regeneration have not been well explored. Moreover, the intracellular Ser/Thr kinase domain located at the carboxyl terminus of the membrane channel is reported to modulate cellular functions by directly modifying channel activity[45,72] or interacting with nuclear chromatin-remodeling complexes[44]. We observed that an increase in extracellular $Mg^{2+}$ contributed to the upregulation of TRPM7 expression, as well as the cleavage and nuclear translocation of TRPM7 kinase fragments (M7CKs). Consequently, in consistent with a previous study[44], these chromatin-bound M7CKs induced the phosphorylation of His-tone H3 at Ser 10 (p-H3S10), which has been reported to be critical for the recruitment of IKK to NF-κB signaling responsive promotors[73] and the binding of IKK to a variety of inflammatory gene promoters, such as CCL5 and IL-8[73–75]. Our ChIP assay demonstrated that $Mg^{2+}$ led to an increased association of p-H3S10 with the promoters of IL-8 and IκBα, thus establishing a role of $Mg^{2+}$ in activating NF-κB signaling and the production of inflammatory cytokines like IL-8.

This study demonstrates the role of TRPM7 in the $Mg^{2+}$-induced activation of the NF-κB signaling pathway, which is crucial in inflammation responses[76]. However, our experiments with the TRPM7 blocker FTY720 revealed that the channel activity of TRPM7 does not account for the entire spectrum of defects seen in TRPM7 siRNA-treated macrophages, especially in terms of the phosphorylation of H3S10 and the activation of NF-κB signaling. This implies the cleaved kinase domain may play a more important role in the regulation of inflammatory cytokines in response to $Mg^{2+}$. Recent evidence suggests that divalent cations like $Ca^{2+}$ may play dual roles in macrophages through TRPM7: a rapid role in jump-starting TLR4 endocytosis and possibly a slower role in tailoring an appropriate inflammatory response by modifying the activity of NF-κB[77]. Therefore, we propose that the TRPM7-mediated inflammatory response to $Mg^{2+}$ can be time-dependent: an initial M7CK mediated phosphorylation of H3S10 located at specific gene promoters followed by a better-tailored activation of NF-κB signaling. This idea is supported by our observation that the promoting effect of $Mg^{2+}$ on IL-8 secretion and the inhibitory effect on IL-1β was abrupted between 24–48 h after the treatment. Moreover, since IL-8 also serves as a strong inducer for the activation of NF-κB signaling, the increased IL-8 in the microenvironment resulting from the M7CK-mediated histone modification can enhance the NF-κB signaling cascade through the IL-8 receptor (CXCR1/2) expressed on the membrane of macrophage[78,79]. This explains why prolonged exposure to $Mg^{2+}$ leads to unnecessary over activation of NF-κB signaling and osteoclastogenesis.

We detected a group of CD68$^+$ and TRAP$^+$ osteoclastic-like cells in the wound, which peaked on day 7 after the delivery of $Mg^{2+}$. Unlike the TRAP$^+$ multinucleated cells observed in the bone remodeling stage, osteoclastic-like cells at an early stage of bone healing have been suggested to represent inflammatory responses to biomaterials instead of bone resorption[80]. Indeed, they are actively involved in bone formation by the secretion of inflammatory cytokines and chemokines[81]. We noticed that a short-term delivery of $Mg^{2+}$ contributed to an increased number of TRAP$^+$ osteoclastic-like cells at the inflammation stage but ended with a lower number of bone-resorbing osteoclasts in the remodeling stage than the control or sham group. By contrast, when the delivery of $Mg^{2+}$ was extended, the inhibitory effect of $Mg^{2+}$ on osteoclastogenesis at the remodeling stage disappeared. Given the above-mentioned influence of $Mg^{2+}$ on the activation of NF-κB signaling, which plays a central role in RANKL-induced osteoclastogenesis[82], the administration of $Mg^{2+}$ beyond the early inflammation stage may be detrimental for bone regeneration.

In addition to the osteoclastogenic effect, we observed that the prolonged exposure to $Mg^{2+}$ could also interfere with biomineralization. While a high level of $Mg^{2+}$ (10×) administered at the beginning of the osteogenic induction of MSC could promote extracellular matrix formation and mineralization, the supplementation of $Mg^{2+}$ at the late stage dramatically impaired the mineralization of MSC. This finding recapitulated our in vivo data that the prolonged $Mg^{2+}$ treatment could reduce BMD, as extensively reported elsewhere[12,52]. The formation of calcified tissue in the biological environment starts with the nucleation and growth of calcium phosphate precursors. The presence of $Mg^{2+}$ may inhibit this process by selectively stabilizing more acidic hydrated precursor phases, like dicalcium phosphate[83,84]. Moreover, the adsorption of excessive $Mg^{2+}$ on the surface of precursor and nuclei could further lead to the blocking of the active growth sites[85]. Thus, this study reveals the biphasic nature of $Mg^{2+}$ effect on biological mineralization: during the early phase of MSC differentiation, $Mg^{2+}$ itself and, to a much larger extent, osteogenic factors secreted by macrophage upon the stimulation of $Mg^{2+}$, effectively promote osteogenic differentiation. However, as MSC is committed to osteoblast lineage, they become less responsive to $Mg^{2+}$ and the $Mg^{2+}$-induced immune factors.

Instead, the persistent presence of $Mg^{2+}$ beyond this stage suppresses mineralization and promotes osteoclastic bone resorption.

This study establishes the diverse and multifaceted roles of $Mg^{2+}$ in bone healing (summarized in Fig. 7i). The delivery of $Mg^{2+}$ contributes to the infiltration and the activation of the $CD68^+$ macrophages in bone defects. The influx of $Mg^{2+}$ through the TRPM7 channel and the nuclear translocation of M7CKs lead to the polarization of the macrophages into a pro-osteogenic subtype facilitating the recruitment and the osteogenic differentiation of MSC, which is distinct from the classical M1 or M2 phenotypes. Our findings suggest that the functional plasticity of macrophages may be more prevalent than previously described, eliciting specific responses tailored for different tissue or injury types. It is possible that the initial immune response to bone injury, orchestrated by cell types in the monocyte-macrophage-preosteoclast lineage, can be tuned to support tissue regeneration by a variety of signals, with $Mg^{2+}$ being a defining example. However, the osteo-promoting functions of $Mg^{2+}$ only operate during the early phase of osteogenesis as the continued stimulation of $Mg^{2+}$ leads to the over-activation of NF-κB signaling. Moreover, the presence of $Mg^{2+}$ and $Mg^{2+}$-induced inflammatory cytokines in the later remodeling stage of bone healing increases the number of $TRAP^+$ cells and inhibits the calcification of the extracellular matrix, resulting in compromised bone formation. In light of our findings, the use of $Mg^{2+}$-based biomaterials as orthopedic implants must be viewed with caution, as these materials usually sustain an extended-release of $Mg^{2+}$ which permeates into the wound with uncontrolled dose kinetics. The delicate balance between the osteo-promoting and osteo-inhibitory effects of $Mg^{2+}$, if unchecked, will lead to unpredictable clinical outcomes.

## Methods

**Preparation of alginate gel**. 20% alginate gel (Alg) was prepared by mixing sodium alginate powder (Sigma-Aldrich, USA) in deionized water, while 10% magnesium chloride ($MgCl_2$, Sigma-Aldrich) was used for the preparation of magnesium cross-linked alginate (Mg-alg). The gel was then loaded in a syringe and sterilized by gamma rays (Co-60) irradiation at a dose of 25 kGy. The as-prepared gel was immersed in 1× Phosphate-buffered saline (PBS) at a ratio of 0.1 g/mL and kept in an incubator at 37 °C for five days. At the designated time point, the $Mg^{2+}$ concentration in the PBS was measured by inductively coupled plasma optical emission spectrometry (ICP-OES, Optima 2100DV, Perkin Elmer, USA).

**Animal surgery**. All the animal procedures were performed in accordance with a protocol approved by the Committee on the Use of Live Animals in Teaching and Research (CULATR, HKU). 6-8-week old Sprague Dawley (SD) rats, weighing 200–250 g, were purchased from Charles River Lab (USA) and maintained in the specific pathogen-free facilities (Lab animal unit, LAU, HKU). After the rats were anesthetized via intraperitoneal injections of ketamine hydrochloride (67 mg/kg; Alfamine, Alfasan International B.V., Holland) and xylazine hydrochloride (6 mg/kg; Alfazyne, Alfasan International B.V.). A tunnel defect, 2 mm in diameter, was prepared at the lateral epicondyle of each rat using a hand driller. Animals were randomly assigned into one of three treatment regimens; (1) injection of $Mg^{2+}$ hydrogel directly after injury; (2) injection at day 7 post-injury and (3) injection at both day 0 and day 7 post-injury. Controls for each treatment regimen were animals injected with an equivalent dose of alginate hydrogel without $Mg^{2+}$. After the layer-by-layer closure of the wound, the rats immediately received subcutaneous injections of terramycin (1 mg/kg) and ketoprofen (0.5 mg/kg).

**Micro-CT analysis**. In order to monitor the healing process and examine new bone formation around the gel, micro-CT scans were conducted at serial time points after the surgery (i.e., 0, 7, 14, 21, 28, and 56 days). After the rats were anesthetized, the grafted site was scanned using a live animal scanning device (SkyScan 1076, Kontich, Belgium). Two phantom contained rods with the standard densities of 0.25 and 0.75 $g/cm^3$ were scanned with each sample for calibration. The resolution was set at 17.3 μm/pixel. Data reconstruction was done using the NRecon software (Skyscan Company), and image processing and analysis were done using CTAn software (Skyscan Company).

**Fluorochrome labeling**. Two fluorochrome labels were used sequentially to evaluate bone regeneration and remodeling within the defects. Calcein green (5 mg/kg,

Sigma-Aldrich) was subcutaneously injected into rat femora one week after the surgery, while xylenol orange (90 mg/kg, Sigma-Aldrich) was injected three weeks after the surgery. The fluorochrome labels were visualized under fluorescence microscopy (Niko ECL IPSE 80i, Japan). The intensity of fluorescence was analyzed by ImageJ software (NIH, USA).

**Histological analysis**. After euthanasia, the femora were collected and fixed in 4% paraformaldehyde solution. For non-decalcified sections, the samples were dehydrated with gradient ethanol and embedded with methyl-methacrylate (MMA, Technovit 9100 New, Heraeus Kulzer, Hanau, Germany) after using xylene as a transition. Then, the embedded samples were cut into slices with a thickness of 200 μm and micro-ground to a thickness of 50–70 μm. The selected sections were stained with Goldner's trichrome (Sigma-Aldrich) staining. For decalcified sections, the samples were decalcified with 12.5% ethylenediaminetetraacetic acid (EDTA, Sigma-Aldrich) for 6 weeks. The specimens were then dehydrated in ethanol, embedded in paraffin, and cut into 5 μm-thick sections using a rotary microtome (RM215, Leica Microsystems, Germany). Haematoxylin and eosin (H&E) staining and TRAP staining (Sigma-Aldrich) was performed on selected sections from each sample following the manufacturer's instructions. Images were captured using polarizing microscopy (Nikon Eclipse VL100POL, Nikon, Tokyo, Japan). The quantification of osteoblasts and osteoclasts in the grafted defect area was done on ×10 histological images using ImageJ software (NIH, USA).

**Scanning electron microscopy-energy dispersive spectroscopy (SEM-EDS)**. MMA embedded bone specimens were sliced and polished to ~100 μm before observation. The elemental composition and distribution in the grafted area were determined by SEM-EDS (SU1510, Hitachi, Tokyo, Japan).

**Immunohistochemistry analysis**. The dewaxed slide was treated with Proteinase K (Sigma-Aldrich) for proteolytic digestion and 3% $H_2O_2$ for the elimination of endogenous peroxidase activity. After blocking with normal goat serum, the slides were incubated with primary antibody overnight at 4 °C, respectively. The primary antibodies used in this study include: rabbit anti-OCN (Abcam), rabbit anti-IL-8 (Abcam), rabbit anti-CCL5 (Abcam), rabbit anti-IL-1β (Abcam), rabbit anti-IL-1ra (Abcam), anti-CD68 (Abcam), rabbit anti-Phospho-Histone H3S10 (Abcam), and rabbit anti-TRPM7 (Abcam). The slides were then incubated with goat anti-rabbit secondary antibody and visualized using Diaminobenzidine (DAB) staining kit (Santa Cruz Biotechnology, Santa Cruz, USA) following the manufacturer's instruction. Immunofluorescent staining was done using Alexa-Fluor 488 conjugated anti-rabbit IgG or Alexa-Fluor 647 conjugated anti-mouse IgG secondary antibodies (Thermo Fisher Scientific) and Hoechst 33324 (Thermo Fisher Scientific). Immunofluorescent images were captured using an LSM 780 confocal microscopy (Zeiss, Germany)

**Detection of inflammatory cytokines**. The femora of the rats grafted with pure alginate or Mg-crosslinked alginate were harvested. The bone tissues in and around the grafted defects, around 1 cm in length, were ground into mud using a ceramic mortar and pestle. The mud of bone tissue was then homogenized in pre-cooled RIPA Lysis and Extraction Buffer (ThermoFisher Scientific) for 1 h. The buffer solution was centrifuged at 15,000 rpm for 20 min at 4 °C. The supernatant was collected for protein concentration quantification with BCA Protein Assay Kit (ThermoFisher Scientific). An equal amount of protein from each sample was subjected to quantitative analysis of IL-8 and IL-1β using an ELISA kit (R&D system) following the manufacturer's instructions.

**Cell culture**. The human monocyte cell line THP1 was obtained from the American Type Culture Collection (ATCC, VA, USA), and maintained in RPMI 1640 supplemented with 10% heat-inactivated fetal bovine serum (FBS, Thermo Fisher Scientific) and 1% (v/v) penicillin/streptomycin (Thermo Fisher Scientific) at 37 °C under 5% humidified $CO_2$. Human mesenchymal stem cells (MSC) was kindly provided by Prof. D. Campana (St Jude Children's Research Hospital, Memphis, Tennessee), and maintained in Dulbecco's modified Eagle medium (DMEM, Gibco) supplemented with 10% FBS (Thermo Fisher Scientific) and 1% (v/v) penicillin/streptomycin (Thermo Fisher Scientific) at 37 °C under 5% humidified $CO_2$. Macrophages were differentiated from THP1 cells using serum-free DMEM supplemented with 10 ng/mL phorbol 12-myristate 13-acetate (PMA, Sigma-Aldrich). After 48 h, the THP1-derived macrophages were further cultured in customized Mg-free DMEM (Thermo Fisher Scientific) supplemented with different concentrations of $Mg^{2+}$ (i.e., 0.08 mM, 0.8 mM, and 8 mM) for another 72 h to allow full differentiation and polarization. Mouse bone marrow macrophages (BMM) were obtained by incubating bone marrow cells isolated from both femurs of C57BL/6J mice in a culture medium supplemented with 20 ng/mL macrophage colony-stimulating factor (M-CSF, R&D) for 6 days[86].

To collect conditional medium (CM), THP1-derived macrophages that have been cultured at different concentrations of $Mg^{2+}$ for three days were washed and incubated with fresh serum-free DMEM (contained 1× $Mg^{2+}$) for 24 h. These media were then collected, mixed 1:2 (v/v) with fresh serum-free DMEM (also contained 1× $Mg^{2+}$) for the culture of MSC. For osteogenic differentiation of MSC, osteogenic

supplements (5 mM β-glycerol phosphate, 0.05 mM L-ascorbic acid 2-phosphate, and $10^{-7}$ M Dexamethasone) were added to the serum-free DMEM. For osteoclastic differentiation of THP1-derived macrophages, RANKL (50 ng/mL), and M-CSF (30 ng/mL) were supplemented in the medium for a 14-day osteoclastic induction.

**Flow cytometry assay**. Flow cytometric assay was performed using FACSCantoII Analyzer (BD Biosciences, USA). Differentiated macrophages were detached with trypsin and washed with 1× PBS. For the detection of macrophage surface markers, cells were incubated with monoclonal mouse anti-human antibodies CD163-Cy5.5, CD206-FITC, and CD80-FITC (BD Biosciences, USA), or relevant isotypes (BD Biosciences) for 1 h at 4 °C in dark. After washing with 1× PBS, the fluorescence was compared to isotypes with 10,0000 events recorded. All flow cytometric data were analyzed using Flowjo software, version 10 (Tree Star, USA).

**Cell attachment and proliferation assay**. The effect of $Mg^{2+}$ on the attachment of THP1-derived macrophages and the proliferation of MSC was assessed using a cell counting kit-8 (CCK-8, Dojindo, Japan). At designated time points, the culture medium was replaced by a fresh serum-free medium containing 10% CCK-8. The optical density (OD) value at the wavelength of 450 nm was measured using a microplate spectrophotometer (SpectraMax 340, Molecular Devices, USA). Cell viability was presented in percentage by comparing the OD value of the tested group to that of the control group.

**ATP assay**. After the stimulation of $Mg^{2+}$, the macrophages were lysed, and the intracellular ATP concentrations were determined using a luminescence ATP detection assay (ATPLite, PerkinElmer, USA) following the manufacturer's instructions.

**Cytokine release**. The cytokines produced by THP1-derived macrophages after the stimulation of different concentrations of $Mg^{2+}$ were determined by Proteome Profiler antibody arrays (R&D System) following the manufacturer's instructions. The concentration of IL-8 and IL-1β was further confirmed by specific ELISA kits (R&D System).

**Alkaline phosphatase (ALP) assay**. The effects of $Mg^{2+}$ or conditional medium from macrophages on the ALP activity of MSC were evaluated using the p-NPP method. At the designated time points, the cells were lysed with 0.2% Triton X-100 at 4 °C for 2 h. The supernatant of the lysis after centrifugation was collected and assayed using an ALP detection kit (Sigma-Aldrich) following the manufacturer's instructions. The total protein content was measured using BCA Protein Assay Kit (ThermoFisher Scientific). The relative ALP activity was normalized to total protein content and expressed as units/g·protein.

**Mineralization assay**. Sirius Red staining was used to study the formation of the extracellular matrix of MSC. At the designated time points, cells were fixed with 75% ethanol, and the collagen was stained with Sirius Red solution (Sigma-Aldrich) for 1 h. Alizarin Red staining was used to study the mineralization of MSC. At the designated time points, cells were fixed with 4% paraformaldehyde, and the mineralization nodules were stained with Alizarin Red solution (Sigma-Aldrich) for 5 min. After a thorough wash with Millipore water, the sample was dried in the air before photo taking.

**Real-time quantitative PCR (RT-qPCR) assay**. The total RNA of the cells was extracted and purified using the RNeasy Plus kit (Qiagen, USA) following the manufacturer's instructions. For the reverse transcript, complementary DNA was synthesized using Takara RT Master Mix (Takara, Japan) following the manufacturer's instructions. The primers used in the RT-qPCR assay were synthesized by Life Technologies (ThemoFisher Scientific) based on sequences retrieved from Primer Bank (http://pga.mgh.harvard.edu/primerbank/, Table S1). SYBR Green Premix Ex Taq (Takara) was used for the amplification and detection of cDNA targets on a StepOne Plus Real-time PCR system (Applied Biosystems, USA). The mean cycle threshold (Ct) value of each target gene was normalized to the housekeeping gene GAPDH. The results were shown in a fold change using the ΔΔCt method.

**Western blotting**. At the designated time points after the treatment, the cells were rinsed with ice-cold PBS and lysed with RIPA Lysis and Extraction Buffer (ThermoFisher Scientific). After centrifugation at 15,000×g for 10 min at 4 °C, the supernatants were collected for measuring the protein concentration with BCA Protein Assay Kit (ThermoFisher Scientific). Cytosolic and nuclear extracts were prepared using NE-PER reagents (ThermoFisher Scientific) following the manufacturer's instructions. A total of 30 μg of protein from each sample were subjected to SDS-PAGE electrophoresis and transferred to PVDF membrane (Merck Millipore, USA). Then the membrane was blocked in 5% w/v bovine serum albumin (BSA, Sigma-Aldrich) and incubated with blocking buffer diluted primary antibodies overnight at 4 °C. The primary antibodies used include rabbit anti-TRPM7

(Abcam), rabbit anti-IL-8 (Abcam), rabbit anti-CCL5 (Abcam), rabbit anti-IL-1β (Abcam), rabbit anti-IL-1ra (Abcam), rabbit anti-OPN (Abcam), mouse anti-ALP (Santa Cruz, USA), rabbit anti-OCN (Abcam), rabbit anti-IKKβ (CST, USA), mouse anti-IKKα (CST), rabbit anti-NF-κB p65 (CST), rabbit anti-Phospho-IκBα (CST), mouse anti-IκBα (CST), rabbit anti-Phospho-Histone H3S10 (Abcam), rabbit anti-Histone H3 (Abcam), rabbit anti-Phospho-JNK (CST), rabbit anti-JNK (CST), mouse anti-NFATc1 (Santa Cruz), and mouse anti-β-actin (CST). The protein bands were visualized by ECL substrate (ThermoFisher Scientific) and exposed under ChemiDoc XRS System (BioRad, USA).

**Immunocytochemistry analysis**. Following the stimulation of DMEM containing different concentrations of $Mg^{2+}$, cells were washed with 1× PBS three times, permeabilized with 0.2% Triton X-100 for 10 min either before or after fixation with 4% paraformaldehyde. The primary antibodies used include anti-TRPM7 (Abcam), anti-NF-κB p65 (CST), and anti-Phospho-H3S10 (Abcam). The secondary antibodies used include Alexa-Fluor 488 conjugated anti-rabbit IgG (Thermo Fisher Scientific), Alexa-Fluor 647 conjugated anti-mouse IgG (Thermo Fisher Scientific). FITC-Phallotoxins (Sigma-Aldrich) and Hoechst 33342 (ThermoFisher Scientific) were used for counterstain. The fluorescent images were captured using a Carl Zeiss LSM 780 confocal microscopy (Carl Zeiss, Germany).

**Detection of Mg uptake in macrophages**. THP1-derived macrophages were incubated in $Mg^{2+}$-free DMEM (ThermoFisher Scientific) supplemented with 4 μM Mag-Fluo-4 (ThermoFisher Scientific) and 1 μM F-127 (ThermoFisher Scientific) for 30 min. After that, cells were washed with $Mg^{2+}$-free DMEM and counterstained with Hoechst 33324. Real-time images were captured using a spinning confocal microscope (Perkin Elmer Instruments, USA) immediately after the addition of 8.0 mM $MgCl_2$ into the $Mg^{2+}$-free DMEM. For visualization of mitochondrial, cells were pretreated with MitoTracer (ThermoFisher Scientific) for 1.5 h before observation.

**Super-resolution microscopy**. THP1-derived macrophages were seeded on coverslips with locking beads. After the treatment, cells were washed with 1× PBS three times, fixed with 4% paraformaldehyde, and permeabilized with 0.2% Triton X-100. Then, the cells were incubated with mouse anti-TRPM7 and rabbit anti-Histone H3 primary antibodies overnight. After thorough washing steps with PBS, they were stained with Alexa-Fluor 647 and Alexa-Fluor 488 conjugated secondary antibody (Thermo Fisher Scientific) before being mounted to a Stochastic Optical Reconstruction Microscopy (STORM, Nanobioimaging Ltd., Hong Kong).

**TRPM7 inhibition and blockage**. For TRPM7 silencing, THP1-derived macrophages were transfected with 10 nM siRNA targeting human TRPM7 (SR310261, OriGene, USA) following the manufacturer's instructions using siTran1.0 (OriGene, USA) as the agent. Cells transfected with nonspecific control siRNA (SR30004, OriGene, USA) were used as the control. siRNA transfection efficiency was verified 72 h after the transfection by western blots. For the inhibition of TRPM7 activity, cells were pretreated with 3 μM FTY720 (Sigma-Aldrich) for 2 h, washed with 1× PBS, and subjected to the other assays.

**Chromatin immunoprecipitation**. After the stimulation, chromatin immunoprecipitation (ChIP) assay was performed using a ChIP kit (Abcam) as described by the manufacturer's instruction. In brief, chromatin from crosslinked macrophages was sheared by sonication (8 of 30 s-on and 30 s-off pulses at high power output, Bioruptor Plus, USA) and incubated overnight with rabbit anti-Phospho-Histone H3S10 (Abcam) at 4 °C. Precipitated DNAs were analyzed by RT-qPCR with SYBR Green Premix Ex Taq (Takara) and primers for human IL-8 promoter (−121 to +61) 5′-GGGCCATCAGTTGCAAATC-3′ and 5′-TTCCTTCCGGTGGTTTCT TC-3′, human IκBα promoters (−316 to −15) 5′-GACGACCCCAATTCAAATC G-3′ and 5′-TCAGGCTCGGGGAATTTCC-3′, as well as a human β-actin promoter (−980 to −915) 5′-TGCACTGTGCGGCGAAGC-3′ and 5′-TCGAGCCAT AAAAGGCAA-3′.

**Statistical analysis**. Each experiment was performed at least three times, and the results were expressed as means ± standard deviations (s.d.). For the data analysis, either one-way analysis of variance (ANOVA) followed by Tukey's multiple-comparison post hoc test or two-sample $t$-test was performed with SPSS ver.13.0 (IBM SPSS, USA). The level of significant difference among groups was defined and noted as $*p < 0.05$ and $**p < 0.01$.

**Reporting summary**. Further information on research design is available in the Nature Research Reporting Summary linked to this article.

## Data availability

All the data supporting the findings of this study are available from the corresponding author on request. Source data are provided with this paper.

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

## Acknowledgements

We acknowledge HKU Li Ka Shing Faculty of Medicine Faculty Core Facility for providing a harmonious working environment. This work was financially supported by the National Key R&D Program of China (R&D#2018YFA0703100), General Research Fund of Hong Kong Research Grant Council (#17214516, #N_HKU725/16), Sanming Project of Medicine in Shenzhen Team of Excellence in Spinal Deformities and Spinal Degeneration Disease (SZSM201612055), Shenzhen Science and Technology Funds (JSGG20180507183242702), Hong Kong Innovation Technology Fund (ITS/287/17 and ITS/405/18), the Science and Technology Commission of Shanghai Municipality (No. 18410760600), International Partnership Program of Chinese Academy of Sciences (GJHZ1850), National Natural Science Foundation of China (81970975), and Guangdong Financial Fund for Hige-Caliber Hospital Construction (174-2018-XMZC-0001-03-2125/D-10). We thank Dr. Stuart Fraser, School of Medical Sciences, University of Sydney, Australia, and Prof. Cao Xu, Department of Orthopedics, School of Medicine, the Johns Hopkins University, for their useful comments.

## Author contributions

W.Q., K.H.M.W., J.S., and W.W. conducted animal surgery and analyzed the results. W. Qiao, J. Wu and J. Li contributed to the cell culture and the in vitro tests. K.H.M. Wong and Z. Lin were responsible for the preparation of Mg-releasing alginate. Z.T. Chen, K. Lai, and Y.W. Lam contributed to the design of in vitro experiments. J.P. Matinlinna, Y. Zheng, S. Wu, and X. Liu provide insightful comments on the material-science-related issues. K.M.C. Cheung and Z.F. Chen contributed to the design of animal models and provided invaluable suggestions about clinical indications of the study. Y.W. Lam, K.W.K. Yeung and Z.F. Chen contributed to data interpretation and supervised the project. W. Qiao illustrated the schematic diagram. W. Qiao, Y.W. Lam and K.W.K. Yeung wrote the manuscript with input from all authors.

## Competing interests

The authors declare no competing interests.
