## [Peer Review File · Nature Communications]

Reviewers' Comments:

Reviewer #1:

Remarks to the Author:

The article investigates the effects of Mg²⁺ alloys in bone regeneration. In vivo experiments show that treatment with Mg²⁺ alloys post lesion leads to better results than if the treatment is delayed. In vivo and ex vivo experiments were performed using standards techniques to observe bone regeneration. Mechanistically, the authors attempted to demonstrate that macrophages play an important role in the bone formation in Mg²⁺ treated animals. In fact, macrophage depleted animals by treatment with clodronate showed reduced bone healing compared to non-treated animals.

This is an interesting study that uses a multidisciplinary approach. However there are a number of aspects that require further consideration.

Specific comments

1. A major concern regarding the in vitro studies is that they are conducted exclusively in THP1 cells, which are a leuakemic line representing a macrophage model. It would be very important to show that the findings in THP1 cells are replicated in primary human or mouse macrophages.
2. Figure 1 The presence of Mg in the group treated with Mg-alg was increased in bones only after 7 days post injury and reduced after 14 days. However, in vitro experiments shown that Mg²⁺ was released up to 28 days (Fig S1a). Is this difference in time course an issue with the preparation or that Mg²⁺ is affected by the in vivo environment?
3. Are there any changes in plasma Mg²⁺ concentration in the in vivo studies?
4. Please provide information about the antibodies for TRPM7. Which antibody was used for the full length and for the kinase? Clone and catalogue number? How many times were experiments performed?
5. All data need to be presented as mean +/- SEM (SD). This is especially relevant for the western blot studies where only represenattves blots are provided.
6. Mg²⁺ increased the TRPM7 full length and cleaved kinase. However how specific is the antibody to be able to detect kinase fragments? There is no information about the antibody, and bands might indicate unspecific bind of the antibody. Is expression of the kinase fragment reduced after SiRNA treatment?
7. The expression of H3S10p in tissues is not very obvious. Positive dots are too weak. Additionally, this experiment show that might have some epigenetic changes, but is does not prove that this happens in macrophages, since there is no colocalization experiments. Please provide better quality data.
8. Figure S4L. SiRNA reduced TRPM7 gene expression in THP1 cells by 40% of. What is the percentage of protein expression after SiRNA?
9. Figure 5i-j: THP1 cells treated with 10xMg show increased IL-8, IL1ra and IL-1b. Similar effects were observed in cells treated with SiRNA TRPM7 in 0.1xMg and 10xMg. These data suggest that TRPM7 reduce the basal levels of IL-8, IL-1ra, and IL1b. If these genes are regulated by TRPM7 and Mg²⁺ in opposite ways, how does extracellular Mg²⁺ increases TRPM7 function and expression?
10. Figure 4L and 4M: Although it is mentioned that SiRNA and FTY are able to reduce NFkB

activation, this was only observed in siRNA cells. However, again is difficult to know, because there is no graph to show reproducibility of the results. Please provide data of multiple experiments

11. Figure 5o,p: results, line 261-264 is confusing. In which experiments, were cells supplemented with RANKL and M-CSF? Gene expression should be accompanied by protein expression as well.

12. Figure 6a : Concerns are raised about the specificity of the proliferation assay used, since it also assesses cell metabolism. Therefore, a more specific proliferation assay is required, such as VFSE or BrDU.

13. Figure 6a-b shows that conditioned medium (CM) from macrophages increased ALP activity and these effects were reduced in low [Mg²⁺] and increased in high [Mg²⁺]. Very similar effects were obtained by simply increasing [Mg²⁺] in medium of MSC. Together these data may indicate that Mg²⁺ itself has effects of the bone formation, without macrophages. These data should be shown in the same figure to allow comparison between groups. As presented, it seems that Mg²⁺ plays a role in bone formation and macrophages have more effect on the proliferation.

14. Figure 6b and S5b show ALP activity in MSC cells incubated with CM from macrophages treated with different [Mg²⁺] and MSC directly treated with different [Mg²⁺]. These data should be in the same graph, since figure S5b is the vehicle control of 5b. Comparing these two figures, they seem to show a similar magnitude of response, indicating that Mg²⁺ has effects on ALP activity and that this is independent on macrophages. These is observed in low and normal [Mg²⁺]. However, at high [Mg²⁺] there is an upregulation of ALP activity in the group treated with CM. Similar effects are observed in gene expression (Fig 6C and 6I). Comparing these two figures, in low [Mg²⁺] there is reduction of osteogenic factors in MSC CM-Mg treated cells, compared to MSC treated with Mg²⁺ only. No differences are observed at normal Mg²⁺ concentration. But CM from [Mg²⁺] treated macrophages induced higher gene expression of osteogenic factors. Please clarify these findings. Also, only mRNA data are presented. What happens to protein expression, which is biologically more relevant?

15. Results Line 297-298 The statement "Instead, most of the late osteogenic markers, such as OPN and BGLAP were not affected by the addition of Mg²⁺" (Fig. 6i) is not entirely correct according to the data shown. Although is difficult to compare because they are in different graphs, all these genes show very similar expression in cells treated with CM-Mg1x (figure 6c) and in Mg1x (6i). Therefore, at normal [Mg²⁺] the participation of soluble factors produced by macrophages do not seem to be important for the gene expression. In addition, OPN and BGLAP are increased in CM-Mgx10, and at low Mg²⁺ concentration, the opposite is observed.

16. These effects are in line with alizarin staining, because figure 6j shows that MSC treated with 1x Mg²⁺ exhibit increased alizarin staining, which reduces in 10xMg, whereas in Figure 6e MSC treated with CM-0.1xMg are less positive for alizarin. All these data together are difficult to explain, since in vivo experiment showed that low Mg²⁺, shown only gene expression.

Minor comments:

1. Add the scale bar of the Y-axis and the complete X-axis in Figure 5b.
2. Add the scale bar of figure S4K. What is the magnification used in the super-resolution microscopy?
3. Figure 5g. Please state that this experiment was done in THP1 cells only
4. Results, Line 245. The figure should indicate 5h and not 6h.
5. Results, line 280: Please check that th text refers to the correct figures. Fig 5g shows IL-8 and IκBα promoters.
6. The flow of figures needs to be improved. The figures should follow numerically in the text.

7. The Introduction and Discussion sections are very long and could be shortened.

Reviewer #2:

Remarks to the Author:

The authors show that Mg²⁺-releasing biomaterials improve bone repair and investigated the underlying mechanisms. The observation that release of Mg²⁺ is only beneficial for bone repair when restricted to the initial phase is interesting and the authors claim a TRPM7-mediated mechanism to explain these findings. Several parts of this complex model are supported by experimental data, but other findings are rather isolated observations without strong connection to the TRPM7-centred model.

1. Effect on osteogenic cells. The authors suggest that Mg²⁺ increases osteogenic differentiation indirectly by stimulating Il-8 secretion by macrophages, that is TRPM7-mediated. However, the authors only show that Il-8 promotes osteogenic differentiation and that Mg²⁺ stimulates Il-8 secretion in macrophages via TRMP7, but do not validate the entire mechanism. To prove this model, the authors should show that the osteogenic effect of the conditioned medium of Mg²⁺-treated macrophages is really mediated by Il-8, by blocking Il-8 presence/activity, and that genetic or pharmacological inactivation of TRMP7 abolishes the osteogenic capacity of the conditioned medium. These experiments are important as the Mg²⁺-induced Il-8 expression in macrophages is not prevented by TRPM7 inhibition, questioning the TRPM7-Il8-osteogenic differentiation-pathway.

2. Effect on macrophages and osteoclasts. Besides an effect of Mg²⁺ on macrophages, the number of (pre-)osteoclasts is also decreased in clodronate mice (Fig. 3C), which the authors consider as being part of the inflammatory response to biomaterials. Moreover, the authors suggest that long-term treatment of macrophages with Mg²⁺ promotes their differentiation to osteoclasts (Fig. 5). To confirm this statement, the effect of inhibiting TRPM7 genetically or pharmacologically on osteoclast formation should be analyzed. In addition, some of the data are rather conflicting: Mg²⁺ seem to increase the number (and activity) of osteoclasts during the early phases of bone healing, but reduces its number at the later stages (Fig. 1). What is the relevance of this temporary effect and how is this regulated.

Mg²⁺ increases the expression of TRPM7 in macrophages, which results in increased intracellular Mg²⁺ levels, according to the discussion (line 387). However, the increase in intracellular Mg²⁺ is already observed after minutes. How to reconcile these data (Fig. 5A,B).

Fig 5 f: the upregulation of M7CK seems very weak, please quantify. Panels g, h: do not fit with the text.

3. The model of bone healing should be better described, as the defect destroys both cortical and trabecular bone, but only the repair of the trabecular bone is described without describing the healing of the cortical bone, which is as important. In addition, the original bone parameters should be included in order to verify that the bone repair results in normal bone structure, a deficit or even excess of bone. Indeed, the bone healing in the Mg-Alg condition (Fig. 1) seems to result in excess bone, as BV/TV is around 30%, which is high and large bone structures without hematopoietic bone marrow are observed (Fig. 1C, D). However, it is difficult to evaluate these parameters when no information on the original bone parameters/structure is given. In addition, no information is given on how the number of osteoblasts and osteoclasts are quantified, what is the region of interest taken, is this the total area of the implant and are cells quantified on all the trabeculae within this region. Finally, the dynamic bone parameters are quantified by measuring the intensity of fluorescence, which is not regarded as the correct parameter to analyze. Instead, double labels, single labels, distance between labels should be measured in order to calculate BFR, MAR,...

4. The quality of the images is too low to distinguish osteoblasts, osteoclasts, macrophages, Il-8,

TRPM7, p-H3S10 and does not allow the reader to detect differences in these parameters as described by the authors. One has mainly to believe what the authors are stating, without having sufficient high-quality immunohistochemistry data.

Minor comment

In all the figures, 'Hochest' should be 'Hoechst'

Fig. 6a. Cell viability is not the same as cell proliferation. It now seems that at day 9 cell viability increases 5 times, indicating that at day most of the cells are dying. Likely, the number of viable cells is quantified, but this is different from cell viability and cell proliferation.

Reviewer #3:

Remarks to the Author:

Thank you for the opportunity to review this paper, entitled "TRPM7 kinase-mediated immunomodulation in macrophage plays a central role in magnesium ion-induced bone regeneration" by W. Qiao, et al.

In this study, the authors have investigated the use of Mg²⁺ impregnated biomaterials as a bone healing strategy following in jury. The authors show that in the early inflammation stage, Mg²⁺ primarily targets CD68 monocyte/macrophage cells and modulates their cytokine and chemokine release. Delivery of Mg²⁺ in the early phase (day 0-7) of bone healing increases bone regeneration, while prolonged delivery has little, and somewhat detrimental, effects on healing. Increased Mg²⁺ levels are shown to upregulate TRPM7 expression in macrophages, leading to increased intracellular levels of Mg²⁺ and cleavage and nuclear translocation of M7CKs. These M7CKs contribute to H3 phosphorylation, which leads to recruitment of IKK to NF-κB signaling responsive promoters and binding of IKK to inflammatory gene promoters, contributing the modulation in immune regulation seen with Mg²⁺ delivery.

This paper is novel and important, given the need for an improved bone healing biomaterial. Mg has been investigated as a biomaterial for bone healing for many years, however the impact of Mg delivery on immune regulation and the optimal Mg delivery strategy have not been well characterized. Overall, this manuscript is complete and the claims are well supported. However, the following points should be considered before final publication:

Major Comments:

1. Figure 3 and pg 7: Text claims a significant increase in CD68+ macrophages after treatment with Mg²⁺ releasing hydrogels. However, the number of CD68+ macrophages are not quantified. Provide additional proof of increase in CD68+ macrophages (IF quantification, flow cytometry of local region, etc.)
2. Pg 7 - "...addition of Mg²⁺ also contributed to a group of macrophage-derived TRAP preosteoclasts at the early stage..." While there are fewer TRAP+ preosteoclasts in the macrophage depleted model, it wasn't shown that these cells are macrophage-derived. Can it be another cell type present in both that macrophages were acting on only in the macrophage-competent model? Please provide additional proof for claiming macrophage-derived.
3. Fig 5f- western blot does not clearly show the pattern claimed.
4. Fig 5 g, h, I, j - please include physiological Mg concentrations (1x) in analyses or provide rationale why 1x Mg concentration is not analyzed, as in other experiments. Are the results with the 10x Mg different than those of the physiological (1x) levels?
5. All figures and data need to be more explicit to state "n" so rigor can be assessed including histology.

Minor Comments:

1. Explain difference between the two y axes in Figure S1.
2. In all figures, correct spelling of "Hoechst".
3. Please check all references to figures 3 and 4 in the text. Some subfigures appear to be

incorrectly referenced.

Reviewer #1 (Remarks to the Author):

The article investigates the effects of Mg²⁺ alloys in bone regeneration. In vivo experiments show that treatment with Mg²⁺ alloys post lesion leads to better results than if the treatment is delayed. In vivo and ex vivo experiments were performed using standards techniques to observe bone regeneration. Mechanistically, the authors attempted to demonstrate that macrophages play an important role in the bone formation in Mg²⁺ treated animals. In fact, macrophage depleted animals by treatment with clodronate showed reduced bone healing compared to non-treated animals.

This is an interesting study that uses a multidisciplinary approach. However there are a number of aspects that require further consideration.

Specific comments

1. A major concern regarding the in vitro studies is that they are conducted exclusively in THP1 cells, which are a leuakemic line representing a macrophage model. It would be very important to show that the findings in THP1 cells are replicated in primary human or mouse macrophages.

We agree with this reviewer on the limitation of making all our *in vitro* observations on a single cell line (THP1). As requested by this reviewer, we have now tested the effect of an elevation of extracellular Mg²⁺ on primary mouse bone marrow macrophages (BMM). We found that the exposure of BMM to 10x Mg²⁺ (8 mM) for 72 hours contributed to similar immunomodulatory effects as observed on THP1-derived macrophages (summarized in the figure below, which is now shown as Fig. 4o in the revised manuscript): genes upregulated by 10x Mg²⁺ included IL-10, OSM, CCL5, and VEGFA, while two major pro-inflammatory genes, TNF- α and IL-1 β , were downregulated. 10x Mg²⁺ also altered the levels of BMP2, TGF- β 1 and IL-6 in BMM in the same trend as in THP1-derived macrophages, although the changes in BMM were not as statistically significant as in THP1-derived macrophages.

O

We reported in our original submission that 10x Mg²⁺ upregulated IL-8 in THP1-derived macrophages. Due to the lack of IL-8 in mice (Singer and Sansonetti, 2004, bibliography is shown in the end of our response to each question), we sought to confirm this finding in mouse BMM by examining the expression of Macrophage Inflammatory Protein 2 (MIP-2), also known as Chemokine (C-X-C motif) ligand 2 (CXCL2), which has been recognized as the murine structural and functional homologue of human IL-8 (Tekamp-Olson et al, 1990, Worgall, 2018). We found that 10x Mg²⁺ treatment led to an ~1.5-fold increase in the expression of MIP-2 compared to the control. Hence, our new data show that an elevated concentration of Mg²⁺ induced comparable changes in the expression of cytokines in primary mouse macrophages and human THP1-derived macrophages (Fig.4f, o). This also corroborates with our observation that similar changes in IL-8 and IL-1β expression were detected in both THP1-derived macrophages and in our animal studies (Fig. 4m,n, S5e,f).

The only discrepancy we observed between the responses of mouse BMM and human THP1-derived macrophages to 10x Mg²⁺ was the expression of IL-1ra: it was found upregulated in THP1-derived macrophages and was downregulated in BMM. This divergence is possibly due to the more pronounced inhibitory effect of Mg²⁺ on the expression of IL-1β and TNF-α in BMM than in THP1-derived macrophages. As IL-1ra expression is known to be influenced by both IL-1β and TNF-α (Sandler et al, 2007), the downregulation of these factors in BMM may in turn suppress IL-1ra expression.

In addition to cytokine expression, we have also confirmed the regulatory effect of Mg^{2+} on TRPM7 in mouse BMM. Similar to our observations on THP1-derived macrophages, 10x Mg^{2+} upregulated the expression of TRPM7, at both protein (Fig. 6a) and RNA (Fig. 6c) levels, the nuclear translocation of M7CKs (Fig. 6d), and the phosphorylation of H3 at residue S10 (Fig. 6e) in mouse BMM. Taken together, these new data on mouse BMM are consistent with the conclusions previously drawn on experiments conducted on THP1 cells, suggesting the Mg^{2+} effects on macrophages are not a phenotype exclusive to the THP1 leukemia cell line. The manuscript has now been revised to include these new data; new changes are highlighted in yellow in the resubmitted document.

Singer, Monique, and Philippe J. and Sansonetti, "IL-8 is a key chemokine regulating neutrophil recruitment in a new mouse model of Shigella-induced colitis." *The Journal of immunology* 173.6 (2004): 4197-4206.

Tekamp-Olson P, Gallegos C, Bauer D, McClain J, Sherry B, Fabre M, van Deventer S, Cerami A et al, : Cloning and characterization of cDNAs for murine macrophage inflammatory protein 2 and its human homologues. *J Exp Med.* 1990, , 172: 911-19

Worgall, S. Stranger in a Strange Land: IL-8 in the Mouse Lung?. *Am J Respir Cell Mol Biol.* 2018;59(5):525-526

Sandler, Charlotta, et al, . "Selective activation of mast cells in rheumatoid synovial tissue results in production of TNF- α , IL-1 β and IL-1Ra." *Inflammation Research* 56.6 (2007): 230-239

2. Figure 1 The presence of Mg in the group treated with Mg-alg was increased in bones only after 7 days post injury and reduced after 14 days. However, in vitro experiments shown that Mg²⁺ was released up to 28 days (Fig S1a). Is this difference in time course an issue with the preparation or that Mg²⁺ is affected by the in vivo environment?

Fig. S1a shows the accumulative Mg²⁺ release from the hydrogel. The concentration of Mg²⁺ in the immersion test stayed virtually unchanged since Day 7, indicating a vast majority of Mg²⁺ had already been released from the hydrogel in the first 7 days. Therefore, the *in vitro* data on Mg²⁺ release are consistent with the *in vivo* finding that the Mg-alginate only contributed to the increase of local Mg²⁺ for no more than 7 days after the surgery.

3. Are there any changes in plasma Mg²⁺ concentration in the in vivo studies?

We have now tested the Mg²⁺ concentration and Ca²⁺ concentration in the blood plasma of rats grafted with either alginate or Mg-alginate on day 7 after the surgery. As shown in Supplementary Fig.1f, there was no significant difference in the plasma Mg²⁺ or Ca²⁺ concentrations. This suggests that a localized change in Mg²⁺ concentration in the bone wound is sufficient in promoting bone healing.

4. Please provide information about the antibodies for TRPM7. Which antibody was used for the full length and for the kinase? Clone and catalogue number? How many times were experiments performed?

The antibodies used for TRPM7 are all from Abcam and the experiments were all performed more than three times. The number of biological replicates used in all experiments is now stated in the figure legends of the revised manuscript. For the western blot, we first used rabbit polyclonal anti-TRPM7 antibody (Cat No: ab135817), which was raised against a synthetic peptide corresponded to human TRPM7 aa 1827-1850 (C terminal). This epitope is similar to that of the antibody (aa 1817-1863) originally used to discover M7CK (Krapivinsky et al, 2014). Furthermore, an image provided by the manufacturer of the antibody we used (<https://www.abcam.com/trpm7-antibody-c-terminal-ab135817.html>) shows the detection of multiple bands in addition to the full-length band located at 212 kDa in WB, as well as a nuclear staining in immunohistochemistry, exactly as what we showed in our manuscript. To be even more scrupulous, we further validated our findings by using another rabbit monoclonal anti-TRPM7 antibody (Cat No: ab109438), raised against aa 1800 to the C-terminus of human TRPM7, which also showed multiple bands at around 130 kDa, 55 kDa, and 35 kDa, similar to that labeled by ab135817. Taken together, it is extremely unlikely that the lower MW bands we reported in this study (Fig. 5e, f, j, 6a, b) were due to cross-reactivity to proteins unrelated to TRPM7.

Krapivinsky, Grigory, et al. "The TRPM7 chanzyme is cleaved to release a chromatin-modifying kinase." *Cell* 157.5 (, 2014): 1061-1072.

5. All data need to be presented as mean +/- SEM (SD). This is especially relevant for the western blot studies where only represenattves blots are provided.

The quantitative data for the western blot were shown as mean +/- SD in the figures. We have now quantitated the signals from all the replicates in all Western blot studies in the revised manuscript.

6. Mg²⁺ increased the TRPM7 full length and cleaved kinase. However how specific is the antibody to be able to detect kinase fragments? There is no information about the antibody, and bands might indicate unspecific bind of the antibody. Is expression of the kinase fragment reduced after SiRNA treatment?

As discussed in our response to Question 4, we have validated the specificity by using two different TRPM7 antibodies that target the C-terminal of TRPM7. Only bands detected by both antibodies are recognized by us as kinase fragments of TRPM7. Moreover, the location of the bands for kinase fragments was consistent with previous studies on TRPM7 kinases (Krapivinsky et al, 2014). As shown in Fig. 5k, the expression of the kinase fragments was reduced by siRNA treatment.

Krapivinsky, Grigory, et al., "The TRPM7 chanzyme is cleaved to release a chromatin-modifying kinase." *Cell* 157.5 (2014): 1061-1072.

7. The expression of H3S10p in tissues is not very obvious. Positive dots are too weak. Additionally, this experiment show that might have some epigenetic changes, but is does not prove that this happens in macrophages, since there is no colocalization experiments. Please provide better quality data.

We now show a higher magnification of the immunofluorescence staining that provides a clearer evidence of the phosphorylation of H3S10 in the defect area. Moreover, double staining of CD68 and p-H3S10 demonstrated the phosphorylation of H3S10 occurred in a subset of CD68+ macrophages (Fig.S6e) Furthermore, we have performed the co-staining of TRPM7 and p-H3S10 in mouse BMM and showed that the addition of Mg^{2+} contributed to the nuclear translocation of TRPM7-specific staining and phosphorylated H3S10 in the same nuclei (Fig. 6d). This is consistent with what we have reported on THP1-derived macrophages in our original submission. Taken together, these new data confirm the Mg^{2+} -dependent increase of H3S10 phosphorylation and nuclear translocation of TRPM7 staining in primary macrophages, in vitro and in vivo. The manuscript has now been revised to include these new data; new changes are highlighted in yellow in the resubmitted document.

8. Figure S4L. SiRNA reduced TRPM7 gene expression in THP1 cells by 40% of. What is the percentage of protein expression after SiRNA?

We have now used Western blot to quantitate the effect of siRNA treatment on TRPM7 protein level. As shown in the figure below (Fig. 5j, l), the intensity of the band located at 212 kDa, which corresponds to full-length TRPM7, was reduced by around 80% in siRNA-treated THP1-derived macrophages.

9. Figure 5i-j: THP1 cells treated with 10xMg show increased IL-8, IL1ra and IL-1b. Similar effects were observed in cells treated with SiRNA TRPM7 in 0.1xMg and 10xMg. These data suggest that TRPM7 reduce the basal levels of IL-8, IL-1ra, and IL1b. If these genes are regulated by TRPM7 and Mg²⁺ in opposite ways, how does extracellular Mg²⁺ increases TRPM7 function and expression?

The effect of TRPM7 knockdown on macrophage functions is complicated. Beside Mg²⁺, TRPM7 is known to permit the entry of physiologically essential divalent metal cations including Ca²⁺, Zn²⁺, Mn²⁺, and Co²⁺, as well as non-essential and often environmentally toxic metals such as Ni²⁺, Cd²⁺, Ba²⁺, and Sr²⁺ (Nadler et al, 2001, Schmitz et al, 2003). Chemical or genetic inhibition of TRPM7 may also collaterally affect the intracellular concentrations of these ions, altering numerous biochemical reactions in which these ions are involved. Indeed, accumulating evidence has shown that the activation and

polarization of monocyte-macrophage are associated with TRPM7 and its kinase in multiple ways (Kaitsuka et al, 2014, Nadolni and Zierler, 2018, Schappe et al, 2018). Therefore, it is not unexpected that the knockdown or chemical inhibition of TRPM7 can affect the baseline expression of multiple genes in THP-1 cells, including the ones we tested in this study. The key discovery of this work, we believe, is that the inhibition of TRPM7 activity can abolish the immunomodulatory effect of Mg^{2+} in macrophages. The manuscript has now been revised to clarify this point; new changes are highlighted in yellow in the resubmitted document.

Nadler, M.J.; Hermosura, M.C.; Inabe, K.; Perraud, A.L.; Zhu, Q.; Stokes, A.J.; Kurosaki, T.; Kinet, J.P.; Penner, R.; Scharenberg, A.M.; et al. . LTRPC7 is a Mg²⁺-ATP-regulated divalent cation channel required for cell viability. *Nature* 2001, 411, 590–595.

Schmitz, C.; Perraud, A.L.; Johnson, C.O.; Inabe, K.; Smith, M.K.; Penner, R.; Kurosaki, T.; Fleig, A.; Scharenberg, A.M et al. . Regulation of vertebrate cellular Mg²⁺ homeostasis by TRPM7. *Cell* 2003, 114, 191–200.

Nadolni, Wiebke, and Susanna Zierler. "The channel-kinase TRPM7 as novel regulator of immune system homeostasis." *Cells* 7.8 (2018): 109.

Kaitsuka, Taku, et al. "Inactivation of TRPM7 kinase activity does not impair its channel function in mice." *Scientific reports* 4 (, 2014): 5718.

Schappe, Michael S., et al. "Chanzyme TRPM7 mediates the Ca²⁺ influx essential for lipopolysaccharide-induced toll-like receptor 4 endocytosis and macrophage activation." *Immunity* 48.1 (, 2018): 59-74.

10. Figure 4L and 4M: Although it is mentioned that SiRNA and FTY are able to reduce NFκB activation, this was only observed in siRNA cells. However, again is difficult to know, because there is no graph to show reproducibility of the results. Please provide data of multiple experiments

The quantitative data for the nuclear NF-κB p65 level and the phosphorylation of IκBα are now provided to demonstrate that both FTY720 treatment and TRPM7 siRNA could impair the activation of NF-κB signaling (Fig. 6g). Although FTY720 did not significantly downregulate the expression of IKK-β, it contributed to lowering IKK-α as

compared to the control. This implies that IKK- α , but not IKK- β , is likely to be the critical kinase facilitating the Mg²⁺-induced activation of NF- κ B signaling. This is consistent with previous findings showing that IKK- α plays a more important role than IKK- β in modifying Histone H3 function in nuclei for the activation of NF- κ B signaling (Yamamoto et al, 2003). The manuscript has now been revised to include these new data; new changes are highlighted in yellow in the resubmitted document.

Yamamoto, Yumi, et al. "Histone H3 phosphorylation by IKK- α is critical for cytokine-induced gene expression." *Nature* 423.6940 (2003): 655-659

11. Figure 5o,p: results, line 261-264 is confusing. In which experiments, were cells supplemented with RANKL and M-CSF? Gene expression should be accompanied by protein expression as well.

The cells were supplemented with RANKL (50 ng/mL) and M-CSF (30 ng/mL). An improved version of the description is now added in the method section of the manuscript. To verify our gene expression data, we determined the phosphorylation of JNK, which is essential in RANKL-induced osteoclastic differentiation (Ikeda et al, 2008). Our data show that, while a depletion of Mg²⁺ (0.1x vs 1x Mg²⁺) decreased JNK phosphorylation, a further increase of Mg²⁺ from 1x to 10x did not lead to a significant activation of JNK (Fig. 6l). Moreover, 10x Mg²⁺ actually led to a suppression of nuclear factor-activated T cells c1 (NFATc1) (Fig. 6l), which plays a crucial role in the terminal maturation of osteoclasts (Takayanagi et al, 2002). This is consistent with our RT-qPCR findings that 10x Mg²⁺ only upregulated early osteoclastic markers (i.e., TRAP, RANK, and M-CSF) but suppressed late osteoclastic markers (i.e., CTSK and CTR) (Fig. 6k). Our observation

implies that the TRAP⁺ multinuclear cells enhanced by Mg²⁺ treatment might represent a group of inflammatory cells responsible for early tissue responses to biomaterials instead of bone resorption, as described elsewhere (Lorenz et al, 2015). The manuscript has now been revised to clarify this point.

Ikeda, Fumiyo, et al., "JNK/c-Jun signaling mediates an anti-apoptotic effect of RANKL in osteoclasts." *Journal of Bone and Mineral Research* 23.6 (2008): 907-914.

Takayanagi, Hiroshi, et al., "Induction and activation of the transcription factor NFATc1 (NFAT2) integrate RANKL signaling in terminal differentiation of osteoclasts." *Developmental cell* 3.6 (2002): 889-901.

Lorenz, Jonas, et al. "TRAP-positive multinucleated giant cells are foreign body giant cells rather than osteoclasts: results from a split-mouth study in humans." *Journal of oral implantology* 41.6 (2015): e257-e266.

12. Figure 6a : Concerns are raised about the specificity of the proliferation assay used, since it also assesses cell metabolism. Therefore, a more specific proliferation assay is required, such as VFSE or BrDU.

We understand that, like all other tetrazolium-based tests, our CCK8 assay essentially measures the activity of NADH-mediated electron transport, not proliferative activity of cells. Notwithstanding, the increase of the number of viable cells measured by this test has been widely recognized as an indication of cell proliferation (e.g., Blocki et al, 2015, Wang et al, 2010, Zhao et al, 2012, Bunpetch et al, 2019). To avoid confusion, we have

now changed the label of the y-axis of Fig.6m and Fig.S7d from “cell viability” to "No. of viable cells (% of control)".

Blocki, Anna , et al. "Microcapsules engineered to support mesenchymal stem cell (MSC) survival and proliferation enable long-term retention of MSCs in infarcted myocardium." *Biomaterials* 53 (, 2015): 12-24.

Wang, Lei, et al,. "Differentiation of human bone marrow mesenchymal stem cells grown in terpolyesters of 3-hydroxyalkanoates scaffolds into nerve cells." *Biomaterials* 31.7 (2010): 1691-1698.

Zhao, Lingzhou, et al . "Effects of micropitted/nanotubular titania topographies on bone mesenchymal stem cell osteogenic differentiation." *Biomaterials* 33.9 (,2012): 2629-2641.

Bunpetch, Varitsara, et al,. "Silicate-based bioceramic scaffolds for dual-lineage regeneration of osteochondral defect." *Biomaterials* 192 (2019): 323-333.

13. Figure 6a-b shows that conditioned medium (CM) from macrophages increased ALP activity and these effects were reduced in low [Mg2+] and increased in high [Mg2+]. Very similar effects were obtained by simply increasing [Mg2+] in medium of MSC. Together these data may indicate that Mg2+ itself has effects of the bone formation, without macrophages. These data should be shown in the same figure to allow comparison between groups. As presented, it seems that Mg2+ plays a role in bone formation and macrophages have more effect on the proliferation.

We appreciate this confusion, which is the basis of this and the next three questions, was caused by our unclear explanation of experimental procedures. We thank this reviewer for pointing this out and have now remedied our lack of clarity. To collect “conditioned media”, THP1-derived macrophages that have been cultured at different concentrations

of Mg^{2+} for three days were washed and incubated with fresh serum-free DMEM (contained 1x Mg^{2+}) for 24 hours. These media were then collected, mixed 1:2 (v/v) with fresh serum-free DMEM (also contained 1x Mg^{2+}), and used as “conditioned media” for the culture of MSC. The reason for supplementing the collected media with fresh DMEM is to prevent any depletion of nutrients by macrophages over the course of the 24-hour culture from affecting the subsequent behavior of MSC. This protocol ensures that the Mg^{2+} levels in 0.1x-CM, 1x-CM, and 10x-CM were the same (0.8 mM), and the observed differences in osteogenic effects of these conditional media were likely caused by differences in soluble cytokines secreted by the macrophages, rather than the magnesium concentration itself.

Since the Mg^{2+} concentration in 0.1x-CM, 1x-CM, and 10x-CM was at 1x Mg level, it was reasonable to compare the “conditional medium treated groups” to “the group treated with 1x Mg medium”. We have now modified the Materials and Methods section to clarify this point.

We did observe a minor effect of Mg^{2+} on osteogenic differentiation of MSC. However, the conditional medium (with contained 1x Mg^{2+}) collected from macrophages previously treated with 10x Mg^{2+} (CM-10x) contributed to a significantly higher ALP activity in MSC than the direct exposure of MSC to DMEM with 10x Mg^{2+} (see Fig. 6n & 7g below, noting the difference in scale on the y-axis). Moreover, CM-10x significantly accelerated the mineralization of MSC, unlike 10x Mg^{2+} , who inhibited the mineralization as shown by Alizarin red staining on day 7. Therefore, we believe the osteogenic effect of the conditional medium of Mg^{2+} -stimulated macrophages is greater in strength and different in characteristics than that of Mg^{2+} itself.

14. Figure 6b and S5b show ALP activity in MSC cells incubated with CM from macrophages treated with different [Mg²⁺] and MSC directly treated with different [Mg²⁺]. These data should be in the same graph, since figure S5b is the vehicle control of 5b. Comparing these two figures, they seem to show a similar magnitude of response, indicating that Mg²⁺ has effects on ALP activity and that this is independent on macrophages. These is observed in low and normal [Mg²⁺]. However, at high [Mg²⁺] there is an upregulation of ALP activity in the group treated with CM. Similar effects are observed in gene expression (Fig 6C and 6I). Comparing these two figures, in low [Mg²⁺] there is reduction of osteogenic factors in MSC CM-Mg treated cells, compared to MSC treated with Mg²⁺ only. No differences are observed at normal Mg²⁺ concentration. But CM from [Mg²⁺] treated macrophages induced higher gene expression of osteogenic factors. Please clarify these findings. Also, only mRNA data are presented. What happens to protein expression, which is biologically more relevant?

Experiments on MSC cells directly treated with different [Mg²⁺] are NOT the vehicle controls for those on MSC incubated with CM from macrophages treated with different [Mg²⁺]. Instead, they are independent sets of experiments, because all the CM contained the same Mg²⁺ concentration (1x) while the media used the direct treatment experiments contained different Mg²⁺ concentrations (0.1x to 10x). Furthermore, CM didn't just contain macrophage secretomes, some of the medium ingredients had been partially consumed by the macrophages during the 24-hour culture. Hence, we believe it is unfair to directly compare the behavior of MSC treated with CM and fresh DMEM (as in direct Mg²⁺ treatment). We acknowledge that this confusion was caused by the lack of clarity in

describing our experimental procedures. We hope this issue is now addressed. Please kindly see our answer to Q.13.

As explained previously, although the CM and direct treatments showed the same trend, their impact on osteogenesis was not at the same level. First, 10x Mg²⁺ only contributed to a minor increase in ALP gene expression in MSC (Fig.7g). Secondly, 10x Mg²⁺ didn't upregulate late osteogenic markers, like OPN and BGLAP (Fig.S7e). Instead, it inhibited the mineralization of extracellular matrix as shown by Alizarin Red staining (Fig. S8a). By contrast, CM-10x, even at a same Mg²⁺ level as 1x Mg, was able to induce more than two-fold increase in ALP activity (Fig. 6n) and more than 1.5-fold increase in a variety of osteogenic gene markers, including ALP, COL1A1, OPN, IBSP, BGLAP and Osterix (Fig. 6o).

To validate our RT-qPCR data, we have now provided representative WB images and the corresponding quantification on the effect of Mg²⁺ and CM on the expression of osteogenic markers in MSC (Fig. S7a, f). Similar to our findings in gene expression analyses, our data suggested there might exist some factors in CM-0.1x that significantly inhibit the osteogenic differentiation of MSC. Meanwhile, CM-10x was able to promote the expression of ALP and OCN in MSC, while the direct treatment 10x Mg²⁺ failed to show this effect. In summary, Mg²⁺ can directly induce osteogenesis in MSC, as suggested by numerous previous studies, however, we discovered that Mg²⁺-induced cytokines from macrophages can simultaneously stimulate in the osteogenic differentiation of MSC, and to a greater magnitude than the direct effect of Mg²⁺.

15. Results Line 297-298 The statement “Instead, most of the late osteogenic markers, such as OPN and BGLAP were not affected by the addition of Mg²⁺’ (Fig. 6i) is not entirely correct according to the data shown. Although is difficult to compare because they are in different graphs, all these genes show very similar expression in cells treated with CM-Mg1x (figure 6c) and in Mg1x (6i). Therefore, at normal [Mg²⁺] the participation of soluble factors produced by macrophages do not seem to be important for the gene expression. In addition, OPN and BGLAP are increased in CM-Mgx10, and at low Mg²⁺ concentration, the opposite is observed.

We do not fully understand the comment that “all these genes show very similar expression in cells treated with CM-Mg1x and in Mg1x”, as the expression levels of all tested genes were normalized as one in these figures. As explained in our answer to Q.14, we believe that CM treatment and direct Mg²⁺ treatment are two independent experiments, and thus used CM-Mg1x and in Mg1x as the internal controls for the respective

experiment. In other words, we did not intend to compare the levels of genes treated with CM-1x vs 1x Mg1x, but to compare the effect of CM-10x vs CM-1x, and Mg10x vs Mg1x. Our data show that Mg10x (as compared to Mg1x) did not appreciably increase the expression of OPN and BGLAP in MSC, while CM-10x (as compared to CM-1x) significantly upregulated these two genes (even as the Mg²⁺ in both CM-10x and CM-1x were the same).

We do notice that CM-0.1x (as compared to CM-1x) downregulated all osteogenic genes in MSC (note that the concentration of Mg²⁺ in CM-0.1x was Mg1x), to a much larger extent than Mg-0.1x (as compared to Mg1x). It is possible that Mg²⁺ deficiency might trigger the secretion of anti-osteogenic factors by macrophages. While this may open a new dimension in the study of the immunomodulatory effect of Mg²⁺, the scope of this paper is limited to the understanding of how a transient and localized increase of Mg²⁺ promotes bone healing.

16. These effects are in line with alizarin staining, because figure 6j shows that MSC treated with 1x Mg²⁺ exhibit increased alizarin staining, which reduces in 10xMg, whereas in Figure 6e MSC treated with CM-0.1xMg are less positive for alizarin. All these data together are difficult to explain, since in vivo experiment showed that low Mg²⁺, shown only gene expression.

These data illustrate one of the most important findings of this study: the time- and concentration-dependent effect of Mg²⁺ on osteogenesis. Inspired by this question, we have now arranged these figures in the revised manuscript (Fig. 7a and 7h) to highlight the novelty of these results. As pointed out by this reviewer, MSC treated with 1x Mg²⁺ (for 14 days) exhibit increased alizarin staining, which was reduced in 10x Mg²⁺ (Fig. 7h). Interestingly, if MSC were treated with 10x Mg²⁺ for 7 days, followed by 1x Mg²⁺ for the next 7 days, the alizarin staining intensity became indistinguishable from cells treated with 1x Mg²⁺ for 14 days. However, cells treated with 1x Mg²⁺ for first 7 days, followed by 10x Mg²⁺ for the next 7 days showed reduced alizarin staining. This means, the inhibitory effect of Mg²⁺ on mineralization is only prominent in the second week. Even at 0.1x Mg²⁺, which suppressed alizarin staining (in consistent with the RT-qPCR data

discussed in Q.15), the exposure to 10x Mg²⁺ during the second week could further reduce alizarin staining.

The effect of CM on alizarin staining of MSC, however, was highly different. Although the concentration of Mg²⁺ in CM-0.1x, CM-1x and CM-10x was the same (equal to 1x Mg²⁺, see Q. 13), exposure to CM-10x significantly increased alizarin staining of MSC, while CM-0.1x decreased alizarin staining (in consistent with our RT-qPCR data) (Fig 7a). Interestingly, if CM-10x was replaced by DMEM (1x Mg²⁺) in the second week of the experiment, alizarin staining intensity was unaffected. But if CM-10x was replaced by DMEM in the first week, alizarin staining was significantly reduced. This means, unlike the direct effect of Mg²⁺, the osteogenic effect of CM-10x was predominately exercised during the first week of the treatment.

Interestingly, the inhibitory effect of CM-0.1x on mineralization can be rescued by replacing the culture medium to normal DMEM. These data collectively demonstrate the direct effect of Mg²⁺ and the indirect effect of Mg²⁺ (through immunomodulation of macrophages) on MSC were distinct. We believe that a clear understanding on the distinct direct/indirect effects of Mg²⁺ at different stages of bone healing may revolutionize the design of Mg²⁺-based or Mg²⁺-modified biomaterials for tissue engineering.

Minor comments:

1. Add the scale bar of the Y-axis and the complete X-axis in Figure 5b.

The Y-axis showing the mean intensity of fluorescence is now added accordingly.

2. Add the scale bar of figure S4K. What is the magnification used in the super-resolution microscopy?

The scale bar in figure S4K indicates 1 μm , it now is added in the figure legend.

3. Figure 5g. Please state that this experiment was done in THP1 cells only

We state that the CHIP assay was done in THP1-derived macrophage in the result of our manuscript.

4. Results, Line 245. The figure should indicate 5h and not 6h.

We have gone through the figure number and revised the incorrectly referred ones.

5. Results, line 280: Please check that th text refers to the correct figures.

Fig 5g shows IL-8 and I κ Ba promoters.

We have gone through the figure number and corrected the errors.

6. The flow of figures needs to be improved. The figures should follow numerically in the text.

We have now rearranged the flow of figures to coordinate it with the manuscript.

7. The Introduction and Discussion sections are very long and could be shortened.

We have tried our best to revise the introduction and discussion sections to make them more concise. The deleted sentences are indicated in the revised manuscript as strikethrough text.

Reviewer #2 (Remarks to the Author):

The authors show that Mg²⁺-releasing biomaterials improve bone repair and investigated the underlying mechanisms. The observation that release of Mg²⁺ is only beneficial for bone repair when restricted to the initial phase is interesting and the authors claim a TRPM7-mediated mechanism to explain these findings. Several parts of this complex model are supported by experimental data, but other findings are rather isolated observations without strong connection to the TRPM7-centred model.

1. Effect on osteogenic cells. The authors suggest that Mg²⁺ increases osteogenic differentiation indirectly by stimulating Il-8 secretion by macrophages, that is TRPM7-mediated. However, the authors only show that Il-8 promotes osteogenic differentiation and that Mg²⁺ stimulates Il-8 secretion in macrophages via TRMP7, but do not validate the entire mechanism. To prove this model, the authors should show that the osteogenic effect of the conditioned medium of Mg²⁺ treated macrophages is really mediated by Il-8, by blocking Il-8 presence/activity, and that genetic or pharmacological inactivation of TRMP7 abolishes the osteogenic capacity of the conditioned medium. These experiments are important as the Mg²⁺-induced Il-8 expression in macrophages is not prevented by TRPM7 inhibition, questioning the TRPM7-Il8-osteogenic differentiation-pathway.

As requested by this reviewer, we have now tested the osteogenic differentiation of MSC cultured in conditioned medium supplemented with and without an IL-8 neutralizing

antibody. Our data showed that the IL-8 neutralizing antibody could abolish the osteogenic effect of CM-10x in MSC, as indicated by ALP activity, osteogenic gene expression and Alizarin Red staining (summarized in the figure below, which is now shown as Fig. 7b and d in the revised manuscript). Meanwhile, IL-8 neutralizing antibody did not significantly affect ALP activity and osteogenic gene expression of MSC cultured in CM-1x. Interestingly, the effect of CM-10x on the mineralization of MSC was only abolished by IL-8 neutralizing antibody when it was added in the first week of culture, but not in the second week of the experiment (Fig. S7c). These data suggest that Mg2+ treated macrophages secreted IL-8, which induced osteogenesis in MSC especially when administrated during the initial phase of differentiation. We have now revised the manuscript to include these new data. All new changes are highlighted in yellow.

We did not test the osteogenic effect of conditional medium from macrophage after genetic or pharmacological inactivation of TRMP7, because as shown in Fig. 5m and n, TRPM7 and FTY720 lead to a collateral fluctuation of a variety of inflammatory cytokines, including IL-8, IL-1β and, TNF-α. As TRPM7 transports many

physiologically essential divalent metal cations, including Ca^{2+} , Zn^{2+} , Mn^{2+} , and Co^{2+} , as well as non-essential and often environmentally toxic metals such as Ni^{2+} , Cd^{2+} , Ba^{2+} , and Sr^{2+} (Nadler et al, 2001, Schmitz et al, 2003, bibliography is shown in the end of each answer), the chemical or genetic inhibition of TRPM7 will likely affect many biochemical reactions in which these ions are involved, many of which influence macrophage functions. Therefore, the conditional medium produced by TRPM7-depleted macrophages does not necessarily reflect the Mg^{2+} insufficiency scenario and may contain changes in numerous other factors that are irrelevant to the mechanism studied here. Our data have shown the inhibition of TRPM7 can abolish the effect of extracellular Mg^{2+} on the production of pro-osteogenic cytokines in macrophages, suggesting that TRPM7 is essential for the transduction of Mg^{2+} into physiological responses.

Nadler, M.J.; Hermosura, M.C.; Inabe, K.; Perraud, A.L.; Zhu, Q.; Stokes, A.J.; Kurosaki, T.; Kinet, J.P.; Penner, R.; Scharenberg, A.M.; et al. LTRPC7 is a Mg.ATP-regulated divalent cation channel required for cell viability. *Nature* 2001, 411, 590–595

Schmitz, C.; Perraud, A.L.; Johnson, C.O.; Inabe, K.; Smith, M.K.; Penner, R.; Kurosaki, T.; Fleig, A.; Scharenberg, A.M. Regulation of vertebrate cellular Mg^{2+} homeostasis by TRPM7. *Cell* 2003, 114, 191–200

2.Effect on macrophages and osteoclasts. Besides an effect of Mg^{2+} on macrophages, the number of (pre-)osteoclasts is also decreased in clodronate mice (Fig. 3C), which the authors consider as being part of the inflammatory response to biomaterials. Moreover, the authors suggest that long-term treatment of macrophages with Mg^{2+} promotes their differentiation to osteoclasts (Fig. 5). To confirm this statement, the effect of inhibiting TRPM7 genetically or pharmacologically on osteoclast formation should be analyzed. In addition, some of the data are rather conflicting: Mg^{2+} seem to increase the number (and activity) of osteoclasts during the early phases of bone healing, but reduces its number at the later stages (Fig. 1). What is the relevance of this temporary effect and how is this regulated.

The effect of both the genetic and pharmacological inhibition of TRPM7 on osteoclastogenesis have been reported previously. For instance, FTY720 is known to prevent bone loss by reducing osteoclastic activity (Ishii et al., 2009). Meanwhile, the knock-down of TRPM7 by siTRPM7 suppresses RANKL-mediated osteoclastogenesis (Yang et al, 2013). These data suggest a positive correlation of TRPM7 activity and osteoclastogenesis. In agreement, we show that a transient exposure to Mg^{2+} , which induced TRPM7 expression, increased the incidence of TRAP⁺ osteoclasts. However, we believe the TRAP⁺ multinuclear cells we observed in the Mg^{2+} -treated bone wounds represents a group of inflammatory cells responsible for the early tissue response to biomaterials instead of bone resorption, as described elsewhere (Lorenz et al, 2015). This is supported by our data that indicate 10x Mg only upregulated the expression of early osteoclastic markers (i.e., TRAP, RANK, and M-CSF) but suppressed late osteoclastic markers (i.e., CTSK and CTR). Moreover, an increase of Mg^{2+} actually led to the suppression of nuclear factor-activated T cells c1 (NFATc1), which plays a crucial role in the terminal maturation of osteoclast (Takayanagi et al, 2002). We also observed that Mg^{2+} inhibited the expression of osteoclastogenic cytokines IL-1 β and IL-6, and upregulated the expression of IL-1ra in macrophages, suggesting Mg^{2+} treatment may contribute an anti-osteoclastogenic microenvironment at the early stage of bone healing.

Mg²⁺ increases the expression of TRPM7 in macrophages, which results in increased intracellular Mg²⁺ levels, according to the discussion (line 387). However, the increase in intracellular Mg²⁺ is already observed after minutes. How to reconcile these data (Fig. 5A,B).

We agree that the sentence “We found that the expression of TRPM7 in macrophages was intriguingly upregulated upon the stimulation of Mg^{2+} , resulting in the increased intracellular level of Mg^{2+} , as well as the cleavage and nuclear translocation of TRPM7 kinase fragments (M7CKs)” is not entirely accurate as the initial entry of Mg^{2+} was not associated with the slower upregulation of TRPM7. Therefore, we have now revised it as follows:

“We observed that an increase in extracellular Mg²⁺ contributed to the upregulation of TRPM7 expression, as well as the cleavage and nuclear translocation of TRPM7 kinase fragments (M7CKs)”

Fig 5 f: the upregulation of M7CK seems very weak, please quantify. Panels g, h: do not fit with the text.

We have now repeated the experiment by increasing the loading volume of proteins and updated the western blot image with clearer bands. Meanwhile, we quantified the intensity of the bands in three experiments and provided the quantitative data.

Ishii, Masaru, et al. "Sphingosine-1-phosphate mobilizes osteoclast precursors and regulates bone homeostasis." *Nature* 458.7237 (2009): 524-528

Yang, Yu-Mi, et al. "TRPM7 is essential for RANKL-induced osteoclastogenesis." *The Korean Journal of Physiology & Pharmacology* 17.1 (2013): 65-71

Lorenz, Jonas, et al. "TRAP-positive multinucleated giant cells are foreign body giant cells rather than osteoclasts: results from a split-mouth study in humans." *Journal of oral implantology* 41.6 (2015): e257-e266

Takayanagi, Hiroshi, et al. "Induction and activation of the transcription factor NFATc1 (NFAT2) integrate RANKL signaling in terminal differentiation of osteoclasts." *Developmental cell* 3.6 (2002): 889-901

3. The model of bone healing should be better described, as the defect destroys both cortical and trabecular bone, but only the repair of the

trabecular bone is described without describing the healing of the cortical bone, which is as important. In addition, the original bone parameters should be included in order to verify that the bone repair results in normal bone structure, a deficit or even excess of bone. Indeed, the bone healing in the Mg-Alg condition (Fig. 1) seems to result in excess bone, as BV/TV is around 30%, which is high and large bone structures without hematopoietic bone marrow are observed (Fig. 1C, D). However, it is difficult to evaluate these parameters when no information on the original bone parameters/structure is given.

In this study, we created a critical defect that cannot heal spontaneously by introducing tunnel defects that went through the femur, destroying both the trabecular bone and the cortical bone at two ends. At each time point after the surgery, we only quantified the newly formed bone tissue in the tunnel defect. As a result, it was difficult to distinguish the trabecular bone from the cortical bone, especially in early time points when the structure of the newly formed bone was irregular.

As suggested by this reviewer, the original bone parameters have now been quantified and the mean value was added to the figure (shown as dash line) for comparison (Fig. 1c). Mg-Alg contributed to excessive bone in the defect, in consistent with previous observations that other Mg²⁺ releasing biomaterials also contributed to excessive new bone formation (Chaya et al, 2015, Zhang et al, 2016). It is worth noting that the quick replenishment of bone tissue without hematopoietic bone marrow is beneficial in some clinical cases. For example, the presence of excessive amount of bone tissue promises the initial stability of dental/orthopaedic implants and prevents the refracture.

In addition, no information is given on how the number of osteoblasts and osteoclasts are quantified, what is the region of interest taken, is this the total area of the implant and are cells quantified on all the trabeculae within this region.

The quantification of osteoblasts and osteoclasts was done on 10x histological image using ImageJ. Only the area of the gel grafted defect was taken as region of interest. The description has been added to the methodology section of the manuscript.

Finally, the dynamic bone parameters are quantified by measuring the intensity of fluorescence, which is not regarded as the correct parameter to

analyze. Instead, double labels, single labels, distance between labels should be measured in order to calculate BFR, MAR,...

We agree that ideally double labeling should allow us to study the bone formation rate (BFR) and the mineral apposition rate (MAR) by measuring the distance between two labels. We have previously used double labeling to quantitate bone healing on non-traumatic animal model, as shown in the figure below (right). However, in this study, the newly formed bone in the defect was extremely irregular (left), and the measurement of spaces between the two intermingled labels became challenging and prone to subjective judgement. Therefore, we selected a more practical and objective way of quantifying the double labeling data. This method has been used elsewhere (Zhang et al, 2016).

Chaya, Amy, et al. "In vivo study of magnesium plate and screw degradation and bone fracture healing." *Acta biomaterialia* 18 (2015): 262-269

Zhang, Yifeng, et al. "Implant-derived magnesium induces local neuronal production of CGRP to improve bone-fracture healing in rats." *Nature medicine* 22.10 (2016): 1160-1169

4. The quality of the images is too low to distinguish osteoblasts, osteoclasts, macrophages, Il-8, TRPM7, p-H3S10 and does not allow the reader to detect differences in these parameters as described by the authors. One has mainly to believe what the authors are stating, without having sufficient high-quality immunohistochemistry data.

In our original submission, we intended to use images at a relatively low magnification to avoid giving an impression that we have “cherry picked” regions that supported our conclusions. We now realize that these overviews may have compromised the resolution

of the immunofluorescence images and prevented readers from properly interpreted our results. Therefore, we have now provided a set of new panels of figures (Fig. 1e, f, 3b, c, g, h, 4m, n) that show both low magnification overviews and higher magnification inserts in the revised version of the manuscript.

Minor comment

In all the figures, ‘Hochest’ should be ‘Hoechst’

The spelling of “Hoechst” in the figures has been revised throughout the manuscript.

Fig. 6a. Cell viability is not the same as cell proliferation. It now seems that at day 9 cell viability increases 5 times, indicating that at day most of the cells are dying. Likely, the number of viable cells is quantified, but this is different from cell viability and cell proliferation.

The label of the y-axis of this figure have been revised from “cell viability” to “No. of viable cells (% of control)” to prevent confusion. In such case, the figure demonstrated that on day 5, 7, and 9, the number of viable cells in CM-10x group was significantly higher than that of the other group. Given the same initial cell seeding density, this data indicates that CM-10x promote the proliferation of MSC.

Reviewer #3 (Remarks to the Author):

Thank you for the opportunity to review this paper, entitled “TRPM7 kinase-mediated immunomodulation in macrophage plays a central role in magnesium ion-induced bone regeneration” by W. Qiao, et al.

In this study, the authors have investigated the use of Mg²⁺ impregnated biomaterials as a bone healing strategy following in jury. The authors show that in the early inflammation stage, Mg²⁺ primarily targets CD68 monocyte/macrophage cells and modulates their cytokine and chemokine release. Delivery of Mg²⁺ in the early phase (day 0-7) of bone healing increases bone regeneration, while prolonged delivery has little, and somewhat detrimental, effects on healing. Increased Mg²⁺ levels are shown to upregulate TRPM7 expression in macrophages, leading to increased intracellular levels of Mg²⁺ and cleavage and nuclear translocation of M7CKs. These M7CKs contribute to H3 phosphorylation, which leads to recruitment of IKK to NF-κB signaling responsive promoters and binding of IKK to inflammatory gene promoters, contributing the modulation in immune regulation seen with Mg²⁺ delivery.

This paper is novel and important, given the need for an improved bone healing biomaterial. Mg has been investigated as a biomaterial for bone healing for many years, however the impact of Mg delivery on immune regulation and the optimal Mg delivery strategy have not been well characterized. Overall, this manuscript is complete and the claims are well supported. However, the following points should be considered before final

publication:

Major Comments:

1. Figure 3 and pg 7: Text claims a significant increase in CD68+ macrophages after treatment with Mg2+ releasing hydrogels. However, the number of CD68+ macrophages are not quantified. Provide additional proof of increase in CD68+ macrophages (IF quantification, flow cytometry of local region, etc.)

The number of CD68⁺ macrophages in Alg and Mg-Alg grafted defect was quantified by immunofluorescent staining (n=5). The data are now shown in Fig. 3b of the revised manuscript.

2. Pg 7 – “...addition of Mg2+ also contributed to a group of macrophage-derived TRAP preosteoclasts at the early stage...” While there are fewer TRAP+ preosteoclasts in the macrophage depleted model, it wasn’t shown that these cells are macrophage-derived. Can it be another cell type present in both that macrophages were acting on only in the macrophage-competent model? Please provide additional proof for claiming macrophage-derived.

Liposome-encapsulated clodronate is known to induce apoptosis in phagocytic cells. It is a well-established method for the depletion of macrophage (Alexander et al, 2011, Schlundt et al, 2018). However, as the monocyte-macrophage lineage is also recognized as the osteoclast precursor, the use of liposome clodronate can significantly reduce the number of TRAP⁺ osteoclastic-like cells in mouse fracture model (Lin and O'Connor,

2017, Davison et al, 2014). Additionally, we showed here that these TRAP⁺ cells also expressed CD68, which is known as a macrophage marker. We have revised the manuscript to clarify this point. All new changes are highlighted in yellow in the resubmitted document.

Lin, Hsuan-Ni, and J. Patrick O'Connor. "Osteoclast depletion with clodronate liposomes delays fracture healing in mice." *Journal of Orthopaedic Research* 35.8 (2017): 1699-1706

Davison, Noel L., et al. "Liposomal clodronate inhibition of osteoclastogenesis and osteoinduction by submicrostructured beta-tricalcium phosphate." *Biomaterials* 35.19 (2014): 5088-5097

3. Fig 5f- western blot does not clearly show the pattern claimed.

We have increased the loading volume of the protein and updated the western blot image with clearer bands. Meanwhile, we quantified the intensity of the bands in three experiments and provided the quantitative data in Fig. 5f.

4. Fig 5 g, h, I, j – please include physiological Mg concentrations (1x) in analyses or provide rationale why 1x Mg concentration is not analyzed, as in other experiments. Are the results with the 10x Mg different than those of the physiological (1x) levels?

It has been reported that the Mg^{2+} supplementation can rescue the loss of cellular function caused by the depletion of TRPM7 (Runnels et al, 2001, Ryazanova et al, 2010). Additionally, other Mg^{2+} transporters (e.g., MagT1) that can compensate the loss of TRPM7 function, either caused by chemical blockage of genetical inhibition. Therefore, comparing 10x Mg to 0.1x Mg allows us to better demonstrate the effect of increased extracellular Mg^{2+} on macrophages, especially when TRPM7 was intentionally suppressed. Since we have demonstrated that there was no significant difference between 0.1x Mg and 1x Mg in terms of gene and protein expression of TRPM7 (Fig. 5c, e), the difference in cellular function of 10x Mg-treated macrophage compared with 0.1x-Mg treated one should also apply to 1x Mg.

Runnels, Loren W., Lixia Yue, and David E. Clapham. "TRP-PLIK, a bifunctional protein with kinase and ion channel activities." *Science* 291.5506 (2001): 1043-1047

Ryazanova, Lillia V., et al. "TRPM7 is essential for Mg 2+ homeostasis in mammals." *Nature communications* 1.1 (2010): 1-9

5. All figures and data need to be more explicit to state "n" so rigor can be assessed including histology.

We have now included the number of replicates in each experiment in the figure legend.

Minor Comments:

1. Explain difference between the two y axes in Figure S1

The volume of Mg-alginate gel used for *in vitro* ion release test is not exactly the same as the volume injected into the bone defect. Therefore, while the left y-axis shows the exact Mg^{2+} concentration determined in the immersion test, the right y-axis offers the calculated Mg^{2+} concentration

2. In all figures, correct spelling of “Hoechst”.

The spelling of “Hoechst” in the figures has been revised throughout the manuscript.

3. Please check all references to figures 3 and 4 in the text. Some subfigures appear to be incorrectly referenced.

We have rearranged the figures and gone through the figure number and revised the incorrectly referred ones in the updated version of manuscript.

Reviewers' Comments:

Reviewer #1:

Remarks to the Author:

All concerns have been adequately addressed. This is an interesting study that has been well conducted.

Minor comment

The authors should normalize the scale bar of the y-axis for all the graphs showing ALP activity.

Figure 6N, the scale is 0-60; Fig 7G, scale 0-30; Fig 7B, scale 0-40; fig 7C, scale 0-150. T

Reviewer #2:

Remarks to the Author:

The authors responded to the comments by performing additional experiments, showing higher magnifications of the images and/or adapting the text where necessary.

By showing now a higher magnification of the morphology of the bone formed in the Mg-Alg condition, one gets the impression that many of the osteocyte lacunae are empty as no nuclei are observed (Fig 1f). The authors show that the mechanical properties are comparable with the control group, using nanoindentation, but information should be provided whether there is a difference in osteocyte lacunae (with or without nuclei) between the different conditions.

Reviewer #3:

Remarks to the Author:

Thank you for the opportunity to review the re-submission "TRPM7 kinase-mediated immunomodulation in macrophage plays a central role in magnesium ion-induced bone regeneration"; NCOMMS-20-15243A.

In general, the authors have adequately addressed all initial comments and the revised manuscript is greatly improved. However, there are some comments that need to be addressed before acceptance.

Figure 7 – For studies with conditioned media, please provide rationale why a group with conditioned media from macrophages not treated with Mg²⁺ was not included or include this group. Although this study does focus on delivery of Mg²⁺, a control group without Mg²⁺ to assess untreated healing seems essential. Similarly, was DMEM without Mg²⁺ supplementation included in assessments of alizarin red staining?

Figure 7a, f, h – If possible, providing images at 1 week (before the change of media type) would improve clarity and data interpretation.

"Hoechst" spelling still needs to be corrected in several places.

Reviewer #1 (Remarks to the Author):

All concerns have been adequately addressed. This is an interesting study that has been well conducted.

Minor comment

The authors should normalize the scale bar of the y-axis for all the graphs showing ALP activity.

Figure 6N, the scale is 0-60; Fig 7G, scale 0-30; Fig 7B, scale 0-40; fig 7C, scale 0-150. T

Thank you for this comment from the reviewer. We originally adjusted the y-axis scales of these figures to highlight the change of ALP activities among different samples within individual experiments. We agree with this reviewer that standardizing the scales of ALP activities will help readers compare the data across experiments. We have now modified these figures so that the scales of the y-axis are normalized. Thanks to this suggestion, the pronounced effect of conditioned media from Mg^{2+} -stimulated macrophages on the ALP activity of MSC, compared to the direct treatment of MSC with Mg^{2+} , is now more clearly shown.

Reviewer #2 (Remarks to the Author):

The authors responded to the comments by performing additional experiments, showing higher magnifications of the images and/or adapting the text where necessary.

By showing now a higher magnification of the morphology of the bone formed in the Mg-Alg condition, one gets the impression that many of the osteocyte lacunae are empty as no nuclei are observed (Fig 1f). The authors show that the mechanical properties are comparable with the control group, using nanoindentation, but information should be provided whether there is a difference in osteocyte lacunae (with or without nuclei) between the different conditions.

We thank this reviewer for this insightful suggestion. We agree this is an important point, as the presence of empty lacunae may indicate the apoptosis of osteocytes, a sign of the “inactive” bone (Weinstein, Robert S., et al. "The pathophysiological sequence of glucocorticoid-induced osteonecrosis of the femoral head in male mice." *Endocrinology* 158.11 (2017): 3817-3831). However, Fig 1f is not necessarily an accurate reflection of the abundance of cells in the lacuna, because of the following reasons: (1) It is relatively difficult to discriminate the injected hydrogel from the bone tissue in the TRAP-stained samples. Fig a below shows two serial sections of the same bone tissue stained by TRAP and H&E respectively. Whilst the hydrogel (orange arrows) and the bone tissue (green arrows) can readily be distinguished on the H&E section (Right), the two structures appear nearly identical on the TRAP section (Left). The cavities in the hydrogel can sometimes be confused with empty lacunae. (2) The relatively weak methyl green counterstain we used for the TRAP stained sections (necessary in order not to mask the TRAP staining) made it difficult to discern nucleated cells in the lacuna. Fig b shows Fig 1f in our submitted manuscript. A magnification of the highlighted region in the top right panel reveals the presence of nucleated cells in lacunae (arrows) that are too lightly stained to be clearly visible in the original figure. (3) TRAP staining involves the treatment of sections with an acidic reagent at 37°C followed by extensive washing (Brunner, Julia S., et al. "Environmental arginine controls multinuclear giant cell metabolism and formation." *Nature communications* 11.1 (2020): 1-15). Some loosely bound lacunal cells might have been washed off.

Therefore, to address this issue directly, we have re-analysed the H&E stained sections from animals treated with Mg-alginate and alginate alone (see Figure c below for representative images) at day 56 after the operation (the same time point as in Fig 1f). We quantitated the number of

lacunae observed in serial sections with (illustrated by green arrows, Fig d) and without (orange arrows) nucleated cells. Out of the ~100 lacunae counted under 10x magnification in three rats in each treatment group, there is no significant in the percentage of nucleated cell-containing lacunae (Fig e). Thus, there is no evidence that Mg^{2+} affected the incidence of empty lacunae during bone healing.

Reviewer #3 (Remarks to the Author):

Thank you for the opportunity to review the re-submission “TRPM7 kinase-mediated immunomodulation in macrophage plays a central role in magnesium ion-induced bone regeneration”; NCOMMS-20-15243A.

In general, the authors have adequately addressed all initial comments and the revised manuscript is greatly improved. However, there are some comments that need to be addressed before acceptance.

Figure 7 – For studies with conditioned media, please provide rationale why a group with conditioned media from macrophages not treated with Mg²⁺ was not included or include this group. Although this study does focus on delivery of Mg²⁺, a control group without Mg²⁺ to assess untreated healing seems essential. Similarly, was DMEM without Mg²⁺ supplementation included in assessments of alizarin red staining?

We thank this reviewer for his/her comments. As Mg²⁺ is an essential micronutrient, it is not possible to culture cells in the absence of Mg²⁺ without affecting cell viability. Standard culture media such as DMEM contain 0.8 mM Mg²⁺. To vary the concentrations of Mg²⁺ (including Mg²⁺-insufficiency) in our experiments, we used a Mg-free culture medium customized for this project by Life Technologies. MgCl₂ was added to this medium in order to achieve the final Mg²⁺ concentrations of 8 mM (10x), 0.8 mM (1x) and 0.08 mM (0.1x). We discovered that 0.08 mM is the minimal Mg²⁺ concentration for the survival of THP1-derived macrophages. When we used Mg-free medium in the PMA-induced macrophage differentiation of THP1, the number of adhered macrophages was too low for subsequent assays. In many cases, even the small number of adhered THP1 cells remained in the dish were completely detached after the withdrawal of PMA. This observation was consistent with a previous report that the complete depletion of Mg²⁺ greatly impaired cell adhesion (Martz, Eric. "Immune T lymphocyte to tumor cell adhesion. Magnesium sufficient, calcium insufficient." *The Journal of cell biology* 84.3 (1980): 584-598). We have now modified the Methods section (Page 23, Line 585) to clarify this point.

Figure 7a, f, h – If possible, providing images at 1 week (before the change of media type) would improve clarity and data interpretation.

The images showing the mineralization of MSC at week one (before the change of media type) are now shown in Fig.S8a for better data interpretation.

“Hoechst” spelling still needs to be corrected in several places.

We apologize for missing these typos. The wrong spelling of “Hoechst” in Fig. 1, 2 and S3 have now been revised accordingly.